# Emx2 regulates hair cell rearrangement but not positional identity within neuromasts

**Sho Ohta[1†], Young Rae Ji[1†], Daniel Martin[2], Doris K Wu[1]\***

[1]National Institute on Deafness and Other Communication Disorders, National Institutes of Health, Bethesda, United States; [2]Genomics and Computational Biology Core, National Institute on Deafness and Other Communication Disorders, National Institutes of Health, Bethesda, United States

**Abstract** Each hair cell (HC) precursor of zebrafish neuromasts divides to form two daughter HCs of opposite hair bundle orientations. Previously, we showed that transcription factor Emx2, expressed in only one of the daughter HCs, generates this bidirectional HC pattern (Jiang et al., 2017). Here, we asked whether Emx2 mediates this effect by changing location of hair bundle establishment or positions of HCs since daughter HCs are known to switch positions with each other. We showed this HC rearrangement, redefined as two processes named Rock and Roll, is required for positional acquisition of HCs. Apical protrusion formation of nascent HCs and planar polarity signaling are both important for the Rock and Roll. Emx2 facilitates Rock and Roll by delaying apical protrusion of its nascent HCs but it does not determine HCs' ultimate positions, indicating that Emx2 mediates bidirectional HC pattern by changing the location where hair bundle is established in HCs.

**\*For correspondence:**
wud@nidcd.nih.gov

[†]These authors contributed equally to this work

## Introduction

All animal organs have a defined shape and pattern that are tailored to their functions. The specific pattern for each organ is generated by individual cells or cell types acquiring a defined position before differentiating, which requires a cascade of signaling and fate determination events. However, the molecular mechanisms that link the sequential developmental events from regional patterning to subsequent differentiation are often illusive because of tissue complexity and the long time-span involved in organ formation (*Edlund and Jessell, 1999*; *Dasen and Jessell, 2009*; *Sanes and Yamagata, 2009*; *Delgado and Lim, 2017*). The neuromast of the lateral line system in zebrafish offers an advantage in this regard since it is a simple organ that develops quickly, yet its function is entirely dependent on the cellular organization of its hair cells (HCs), thus making this organ conducive for deciphering how a rudimentary tissue transitions to a functional unit molecularly (*Figure 1A*; *Chitnis et al., 2012*; *Ghysen and Dambly-Chaudière, 2004*; *Ghysen and Dambly-Chaudière, 2007*).

Each neuromast detects water flow in two directions. This property is dependent on the orientation of the stereociliary bundle (also known as the hair bundle) erected on the apical surface of its HC (*Figure 1A*). The hair bundle consists of a number of stereocilia arranged in a staircase pattern that are tethered to the kinocilium (*Kindt et al., 2012*). Deflection of the hair bundle toward its kinocilium opens the mechanotransduction channels on the tips of the stereocilia and results in activation of the HC, whereas deflection of the hair bundle away from its kinocilium results in HC inactivation. Thus, hair bundles in neuromasts are bidirectionally oriented, aligned along either the anterior-posterior (A-P) or dorsal-ventral (D-V) axis of the body (*Figure 1A*; *Flock and Wersäll, 1962*; *López-Schier et al., 2004*).

The mechanism for generating the bidirectional orientation pattern of HCs in the neuromast is not clear but it is attributed to a HC rearrangement process that involves lateral inhibition mediated by Notch signaling (*Jacobo et al., 2019*; *Kozak et al., 2020*). As a HC precursor divides to form two daughter HCs in the neuromast, they undergo rearrangement and switch positions with each other approximately 50% of the time (*Wibowo et al., 2011*; *Mirkovic et al., 2012*), prior to differentiation and hair bundle establishment (*Figure 1B*). Despite the fact that not all HC pairs undergo rearrangement, daughter HCs with hair bundle pointing toward the posterior (A→P) are always located in the

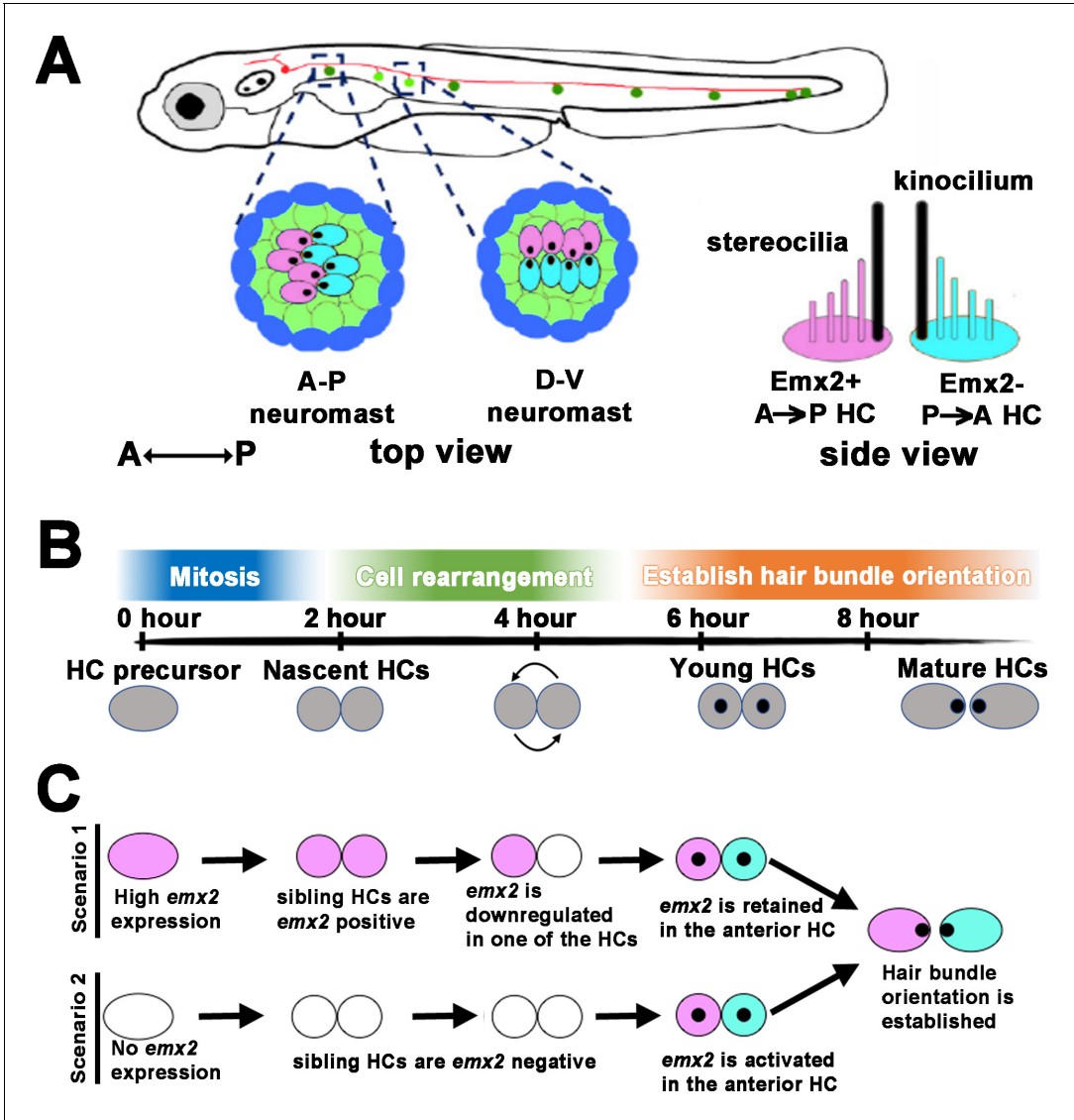

**Figure 1.** Schematic of bidirectional hair cell (HC) establishment in the zebrafish neuromast. (**A**) The lateral line system in zebrafish showing the top view of an anterior-posterior (A-P) and dorsal-ventral (D-V) neuromast, based on their hair bundle orientation. Side view of two hair bundles, each comprises a kinocilium (black) and a stereociliary staircase (pink or blue), arranged in opposite orientation atop of an Emx2-positive (pink) and Emx2-negative (blue) HC. (**B**) An approximate developmental time-line of HC formation in zebrafish neuromast. A HC precursor (gray) divides to form two daughter HCs, which roll to exchange positions with each other 50% of the time before differentiating into mature HCs with opposite hair bundle orientation. The entire process takes approximately 8 hr from detectable Gfp signal driven by the *myosin6b* promoter to HCs with polarized hair bundles based on our live-imaging analyses. (**C**) Emx2, which is important for establishing the bidirectional HC pattern, can be expressed early in the HC precursors, which could affect the HC rearrangement (scenario 1) or later during hair bundle establishment after HCs are formed (scenario 2). A combination of both scenarios is also possible (not shown). Black dots represent the position of the kinocilium.

anterior position of an A-P neuromast (*López-Schier et al., 2004*). Likewise, HCs with hair bundles that point ventrally (D→V) are located dorsally in a D-V neuromast (*Figure 1A*). The hypothesis that HC rearrangement is required for HCs to acquire their positions within a neuromast, though attractive, has not been tested experimentally.

Previously, we showed that the bidirectionally oriented HCs within a neuromast is dependent on the transcription factor Emx2. Emx2 is expressed in only one of the siblings within a HC pair (A→P or D→V HCs), in which the hair bundle is oriented at approximately 180° opposite from the Emx2-negative HCs. As a result, the two sibling HCs show opposite hair bundle orientation (*Figure 1A*; *Jiang et al., 2017*; *Ji et al., 2018*). *Emx2* loss of function (LOF) results in neuromasts with unidirectional hair bundles, P→A and V→D in A-P and D-V neuromasts, respectively. In contrast, e*mx2* gain of function (GOF) neuromasts show only hair bundles in A→P or D→V direction. These results indicate that Emx2 is necessary and sufficient to change hair bundle orientation pattern. However, the timing and mechanism of Emx2's requirement in this process are not clear. For example, Emx2 could be activated early in the HC precursor stage prior to HC formation (*Kozak et al., 2020*) and be involved in HCs' positional acquisition (*Figure 1C*, scenario 1). Alternatively, it could be activated only in one of the sibling HCs (*Jacobo et al., 2019*) and generates the opposite hair bundle orientation after HCs acquired their positions (scenario 2). Either scenario could generate two sibling HCs with opposite bundle orientation pattern (*Figure 1C*). The implication of the role and regulation of Emx2 under each scenario, however, is different. In the former case, Emx2 needs to be downregulated in one of the sibling HCs, whereas in the latter, Emx2 is preferentially activated in one of the sibling HCs to mediate the opposite hair bundle orientation (*Figure 1C*). These two scenarios are not mutually exclusive and a combination of the two scenarios is possible as well.

To distinguish between the two scenarios illustrated in *Figure 1C*, we identified the onset of Emx2 expression using single-cell RNA seq (scRNA-seq) and investigated HC rearrangement in *emx2* GOF and LOF mutants. We found that the HC rearrangement process could be divided into two phases, a Rock phase, when two sibling HCs rock against each other, and a Roll phase, when HCs switch positions with each other. We also tracked the Emx2-positive HCs during the Rock and Roll process using an *emx2* reporter fish, in which *Gfp* is knocked into the *emx2* locus. Our results show that *emx2* transcripts are detectable as HC precursors transition to nascent HCs and Emx2 functions in the Rock and Roll process prior to hair bundle establishment. Emx2 facilitates the Rock and Roll process by regulating apical protrusion formation of the nascent HCs without impairing the final position of HCs within the neuromast. Thus, we concluded that Emx2's key function is to change hair bundle orientation within HCs. Blocking apical protrusion with microtubule inhibitor, nocodazole, or the lack of a core planar cell polarity (cPCP) protein, Van gogh like 2 (Vangl2), affected the Rock and Roll process and positional acquisition of HCs. These results indicate that both apical protrusion formation of nascent HCs and the cPCP pathway are essential for HCs to acquire their proper locations to generate the bidirectional orientation.

## Results

### *emx2* transcription initiates during precursor to daughter HCs transition

To address the role of Emx2 in neuromast HCs, we first investigated the onset of Emx2 expression in the developing neuromast. Emx2 immunoreactivity was first detected in immature HCs before hair bundles are polarized (*Figure 2*, #3) and not in HC precursors or nascent HCs (*Figure 2*, #1 and #2), similar to a previous report (*Jacobo et al., 2019*). However, immunostaining may not be sensitive enough to detect onset of *emx2* transcription. Indeed, a recent study demonstrated enriched Emx2 transcripts and immunoreactivity in HC precursors using immunostaining, in situ hybridization, and scRNA-seq (*Kozak et al., 2020*). These results imply that the downregulation of Emx2 in one of the nascent HCs generated the asymmetry within a pair of HCs (*Figure 1C*). To validate these results, we investigated *emx2* transcription within the neuromast HC lineage by analyzing our existing scRNA-seq data (not published previously) generated from fluorescence-activated cell sorting (FACS) of neuromast HCs using *myo6:ribotag-Gfp* larvae (*Matern et al., 2018*) at four dpf (days post fertilization).

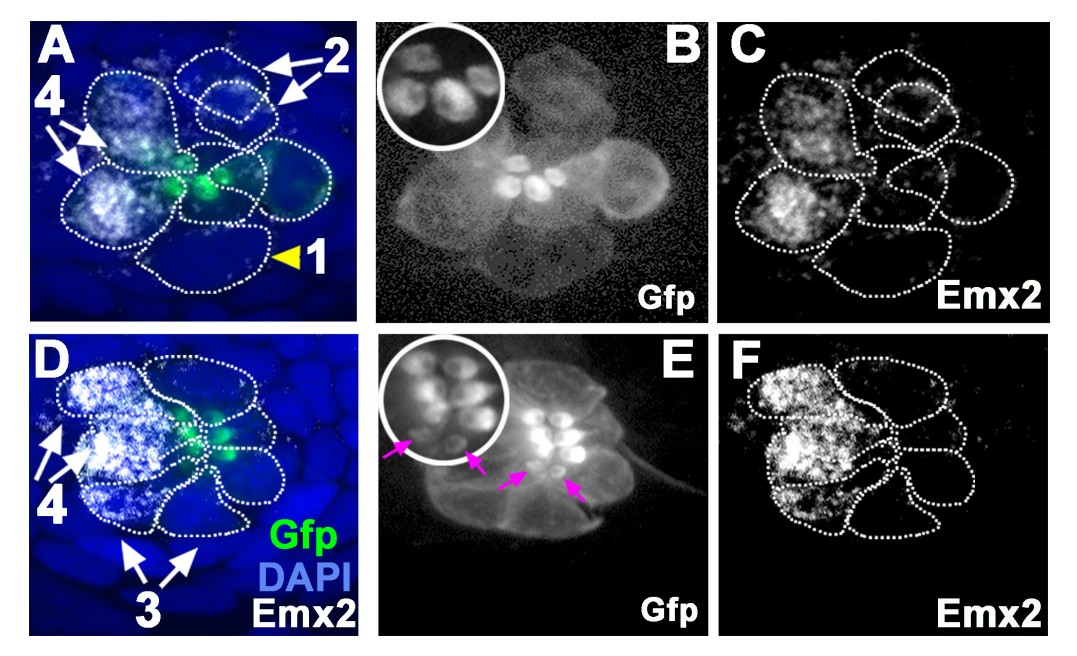

**Figure 2.** Emx2 immunostaining in wild-type (WT) (*myo6b:actb1-Gfp*) neuromasts. (**A and D**) Merged image of DAPI, (**B and E**) β-actin-GFP, and (**C and F**) anti-Emx2 staining of neuromasts. Emx2 immunoreactivity is not detected in dividing hair cell (HC) precursor (#1, arrowhead) or daughter HCs during Rock and Roll (#2). However, it is detectable in immature daughter HCs after Rock and Roll, in which the oriented hair bundle is not yet evident (#3, **E**, magenta arrows), as well as in mature HCs (#4). Both immature (#3) and mature HCs (#4, A→P) that are located in the anterior side of the neuromast are Emx2-positive. Insets in **B** and **E** are higher magnifications showing the hair bundle orientation pattern at the apex of HCs.

Standard quality control procedures were applied to isolate only viable cells, and 2882 of selected cells were used for unbiased clustering. Six major clusters were obtained (*Figure 3—figure supplement 1A*). Based on expression of the transgene (tg_rpl10_3x_HA_short) and selected cell marker genes (*Figure 3—figure supplement 1B and C*) modeled after similar studies (*Lush et al., 2019*; *Kozak et al., 2020*), 2188 cells within the HC lineage from clusters #0, 1, 3 and 4 were selected for reclustering, excluding a supporting cell cluster (Cluster #5) and a cluster with low levels of the transgene (Cluster #2). Reclustering of HC lineage cells resulted in five clusters (*Figure 3A*, *Figure 3—figure supplement 1D and E*). The top 30 marker genes in each cluster were used to assign identities based on expression of known transcripts such as *atoh1a* (enriched in HC precursors and nascent HCs) (*Sarrazin et al., 2006*; *Kidwell et al., 2018*), *myo6b* (expressed in entire HC lineage population but low levels in stressed HCs) (*Kappler et al., 2004*; *Seiler et al., 2004*), and *pvalb8*, *tmc2b*, and *tekt3* (enriched in mature HC) (*Hsiao et al., 2002*; *Maeda et al., 2014*; *Erickson and Nicolson, 2015*; *Figure 3B*). Supporting cell markers *sox2*, *klf17*, *sost*, *stm*, and *si: dkey205h13.2* were expressed in low levels in these clusters (*Figure 3B*; *Hernández et al., 2007*; *Aman and Piotrowski, 2008*; *Thomas and Raible, 2019*; *Söllner et al., 2003*; *Lush et al., 2019*). High ribosomal gene expression was found in HC precursors and nascent HCs, but it was severely downregulated in mature HCs as reported (*Lush et al., 2019*).

Plotting *emx2*-positive cells on UMAP (uniform manifold approximation and projection) showed that they were enriched in the nascent HC population (*Figure 3A and C*, cluster #3) but were only found in a subpopulation of the precursor population (Cluster #4). In contrast, *atoh1*-positive cells were being highest in the precursor population. To investigate the spatial-temporal dynamics of *emx2* expression within the HC lineage, we performed diffusion pseudotime (DPT) analysis. DPT analysis of HC differentiation with heat map representation showed that expression of *emx2* was found during the transition period from HC precursors to nascent HCs, which lagged behind the onset of *atoh1a* and *myo6b* (*Figure 3D*). Consistent with the DPT analysis, the nascent HC cluster showed significantly higher number of *emx2* read-counts (*Figure 3E*) and percentages of *emx2*-positive cells (*Figure 3F*) than other clusters. Taken together, our scRNA-seq results showed that, similar

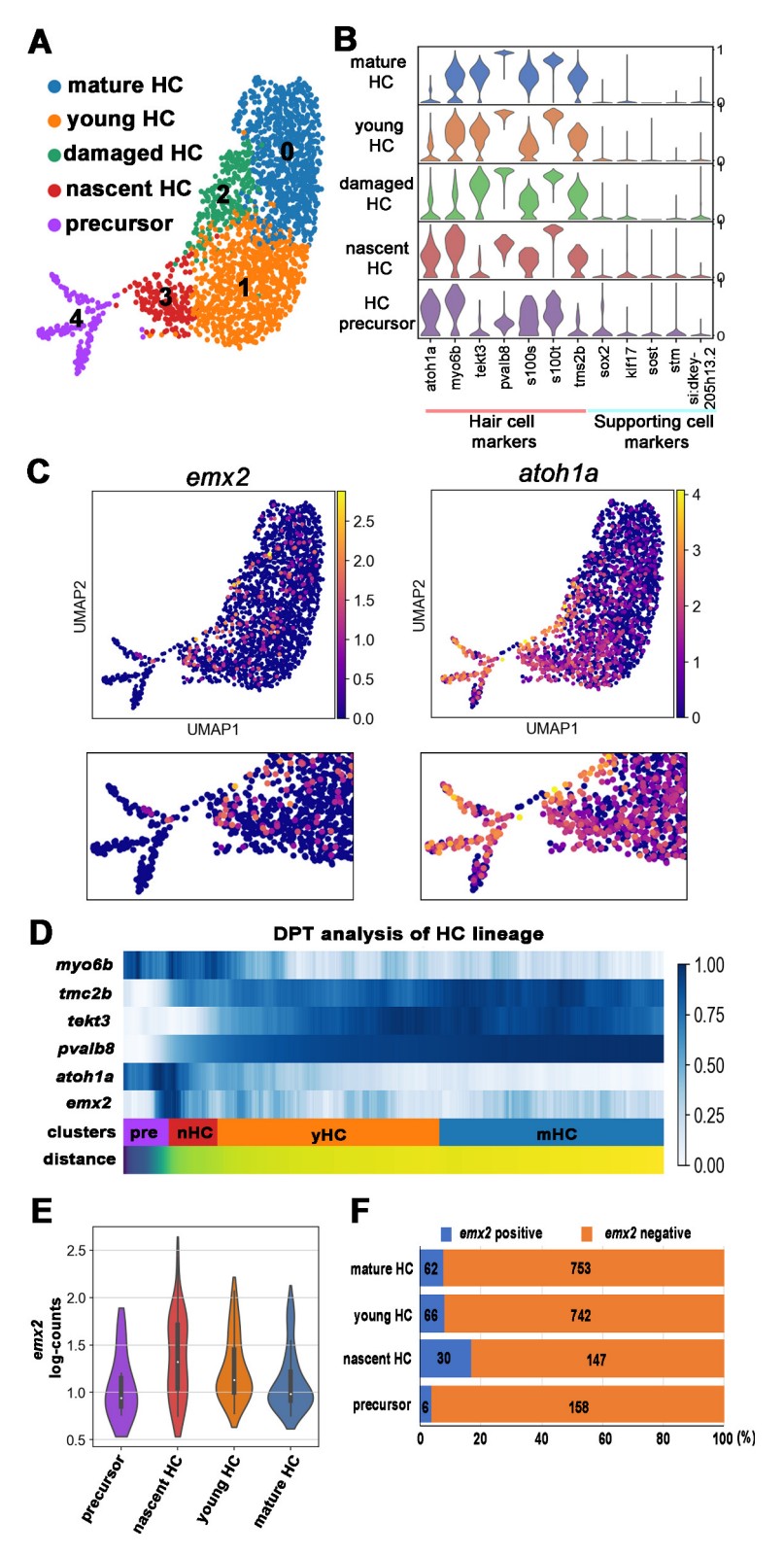

**Figure 3.** scRNA-seq analysis of the hair cell (HC) lineage in neuromasts. (**A**) Uniform manifold approximation and projection (UMAP) plot colored by Louvain clustering annotated with cell types in the HC lineage. (**B**) Violin plots of selected HC (red bar) and supporting cell (blue bar) marker genes in each cluster. (**C**) Distribution of cells expressing *emx2* and *atoh1a* on UMAP. Clusters of precursors (#4) and nascent HCs (#3) are enlarged in bottom panels. (**D**) Heat map representation of selected gene expression dynamics in pseudotime. (**E**) A violin plot showing significantly higher read counts of

*Figure 3 continued on next page*

Figure 3 continued

emx2 in the nascent HC cluster than other clusters (p=0.0069, two tail Student's t-test, source data 1). (F) A bar graph showing a significantly higher number of emx2-positive cells in the nascent HC cluster than others ($X^2$ = 22.44, df = 3, p<0.0001, source data 2). A post-hoc 2 × 2 chi-squared test was also performed for multiple comparisons: precursor HC cluster vs nascent HC cluster, $X^2$ (df = 1)=15.92, p<0.0001; nascent HC cluster vs young HC cluster, $X^2$ (df = 1)=12.73, p=0.0004; nascent HC cluster vs mature HC, $X^2$ (df = 1)=15.08, p<0.0001. The following figure supplements are available for *Figure 3—figure supplement 1*. Identification and annotation of cell types from Louvain clustering.

The online version of this article includes the following source data and figure supplement(s) for figure 3:

**Source data 1.** Comparison of the emx2 log-counts in nascent hair cell (HC) cluster versus other clusters.
**Source data 2.** Comparison of emx2-positive and -negative cells among four hair cell (HC) clusters.
**Figure supplement 1.** Identification and annotation of cell types from Louvain clustering.

to previous results (*Kozak et al., 2020*), emx2 transcripts were found within the HC lineage except we found that the transcripts initiate as HCs transitioned from precursors to nascent HCs rather than peaking early in the precursor cell population as described previously (*Kozak et al., 2020*). The difference between these results is not clear and may be attributed to the difference in the age of the larvae and the number of HCs analyzed.

## The HC rearrangement process is regulated by Emx2

The detection of emx2 transcripts as HC precursors transition to nascent HCs coincides with the reported timing of HC rearrangement (*Wibowo et al., 2011*; *Mirkovic et al., 2012*). To address whether Emx2 affects HC rearrangement, we first live-imaged this process in neuromast HCs, in which β-actin-GFP expression is driven by myo6 promoter (wild-type, WT; *Kindt et al., 2012*). In WT neuromasts, Gfp is first evident in the HC precursor and this expression becomes more prominent as the two daughter HCs undergo the previously described cell rearrangement (*Figure 4A*; *Wibowo et al., 2011*; *Mirkovic et al., 2012*). We found that this cell rearrangement process could be divided into a Rock and a Roll phase (*Figure 4A,A'*, see Materials and methods). Soon after the HC precursor divided into two daughter cells, they started a repetitive rocking motion with each other (*Figure 4A,A'*, *Figure 4—video 1*, Rock phase). This was defined as the Rock phase with a duration of 90 min on average (*Figure 4E*). Then, 57% of HC pairs underwent a Roll phase with an average duration of 75 min, to exchange positions with each other (*Figure 4A, A', D, and F*, Roll phase, *Figure 4—video 1*), whereas 40% of HC pairs only rocked but did not roll to exchange positions (*Figure 4D*, No Roll, *Figure 4—video 2*). A minority of HC pairs (3%), though underwent both Rock and Roll, the two daughter HCs rolled back to their original positions at the end of the Roll phase (*Figure 4D*, Roll back).

Compared to WT, the Rock phase of nascent HC pairs in emx2-/- (emx2 LOF) mutants (*Jiang et al., 2017*), which do not express Emx2 (*Figure 4—figure supplement 1*), was significantly shorter for an average of 67 min (*Figure 4B and E*, *Figure 4—video 3*) whereas the duration of the Roll phase was not significantly different (*Figure 4B and F*, *Figure 4—video 3*). The frequency of HC pairs that underwent the Roll phase was also higher (*Figure 4D*, 76% [(59% + 17%)] in emx2 LOF versus 60% [(57% + 3%)] in WT). However, among HC cell pairs that rolled, there was a significant increase in the frequency of pairs that underwent Roll back (*Figure 4D*, Roll back, 17% of emx2 LOF versus 3% of WT). Therefore, the total percentages of HC pairs that switched positions at the end of the Roll phase was similar between emx2 LOF and WT HCs (*Figure 4D*, 59% vs 57%).

In *myo6b:emx2-p2a-nls-mCherry* (emx2 GOF) neuromasts (*Jiang et al., 2017*), in which both nascent HCs are Emx2 positive based on mCherry expression (*Figure 4—figure supplement 1*), sibling HCs spent a significantly longer time in the Roll but not the Rock phase, compared to WT (*Figure 4E and F*). Only 32% of emx2 GOF HC pairs underwent the Roll phase and none rolled back (*Figure 4C and D*, Emx2 GOF, *Figure 4—video 4*), compared to the 57% that switched positions in WT. In short, these results indicate that emx2 LOF HC pairs spent a shorter time in Rock and Roll but showed an increase in Roll frequency, even though some rolled back (*Figure 4G*). In contrast, emx2 GOF HCs showed the opposite results of a longer duration of Rock and Roll and a decrease in Roll frequencies. Cumulatively, this genetic analysis indicates that Emx2 regulates the cell rearrangement process. This regulation is unlikely to be dependent on a differential level of Emx2 between two siblings since neither HC in emx2 LOF pairs expresses Emx2 (*Figure 4—figure supplement 1*).

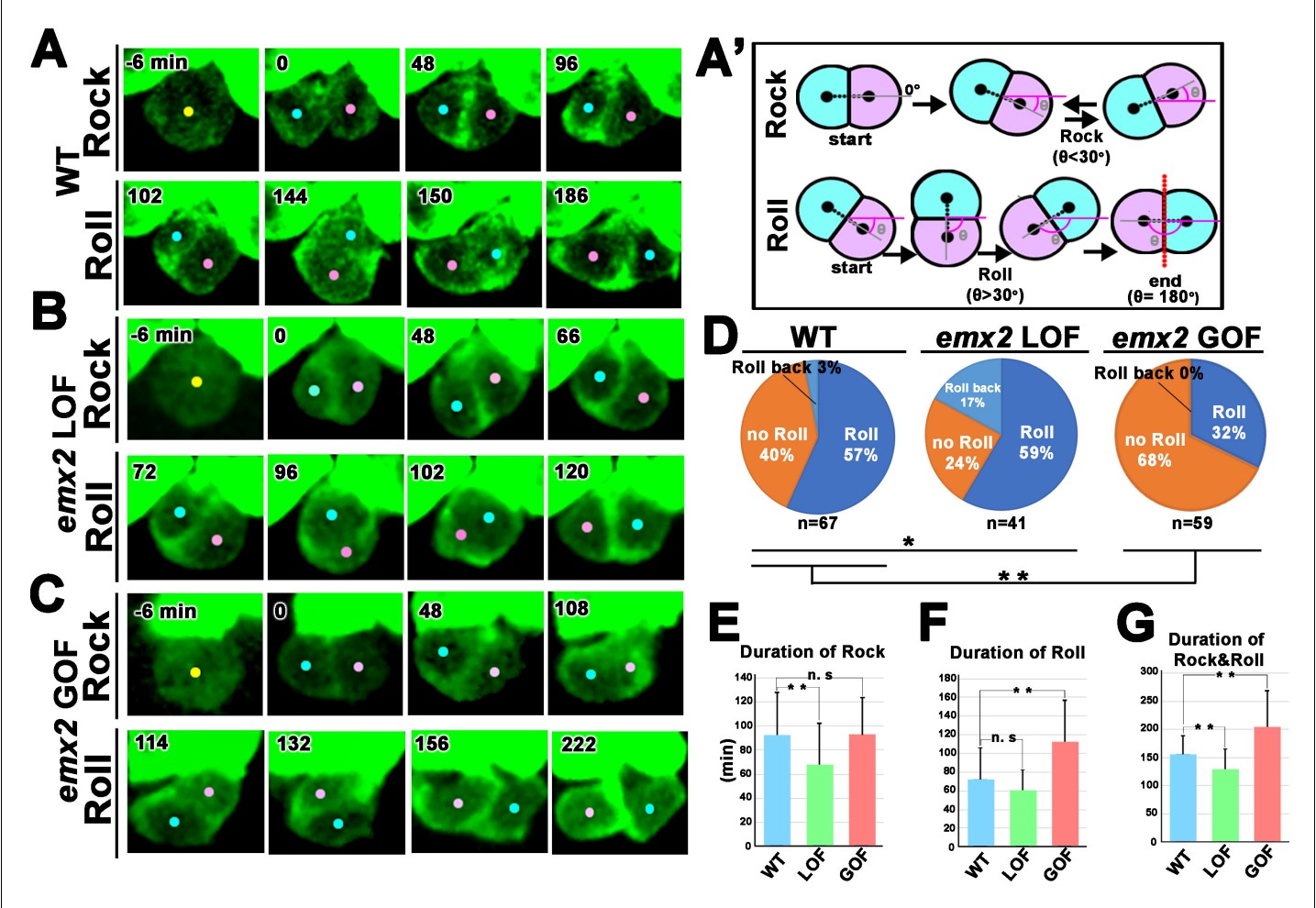

**Figure 4.** Quantification of frequency and duration of Rock and Roll among hair cells (HCs) in wild-type (WT) and *emx2* mutants. (**A**) In WT, a HC precursor (yellow dot) divides to form two daughter HCs (pink and blue dots), which first rock (Rock phase) and then frequently roll to exchange positions with each other (Roll phase). (**A'**) Definition of the Rock and Roll phase (see Materials and methods). (**B and C**) Compared to WT, *emx2* loss of function (LOF) (**B**) and gain of function (GOF) (**C**) HC pairs took a shorter time in Rock and Roll and a longer time in the Roll phase, respectively. (**D**) Frequencies of Roll, No Roll, and Roll back of nascent HC pairs in WT (n = 67 from seven larvae), *Emx2* LOF (n = 41 from six larvae), and GOF (n = 59 from seven larvae). Significance was assessed by using chi-squared test with 3 × 3 contingency table ($X^2$ (df = 4)=30.00, p<0.0001, source data 1). A post-hoc 2 × 3 chi-squared test was performed for multiple comparisons. WT vs. LOF; ($X^2$ (df = 2)=7.95, p=0.018), WT vs. GOF; ($X^2$ (df = 2)=10.39, p=0.0055). Post-hoc chi-squared tests were performed for pairwise comparisons with FDR correction. (**E–G**) Duration of Rock (**E**), Roll (**F**), and Rock and Roll (**G**) of sibling HCs that underwent Rock and Roll in WT (n = 42), *emx2* LOF (n = 35), and GOF (n = 19) larvae. Significance was assessed by using MANOVA (Rock, F(df1 = 2, df2 = 93)=5.349, p=0.0063, Wilks' λ = 0.897; Roll, F(df1 = 2, df2 = 93)=15.638, p<0.0001, Wilks' λ = 0.748; Rock and Roll, F (df1 = 2, df2 = 93)=20.10, p<0.0001, Wilks' λ = 0.698), with post-hoc Tukey's test for pairwise comparisons. *p<0.05, **p<0.01, n.s., not significant. The online version of this article includes the following video, source data, and figure supplement(s) for figure 4:

**Source data 1.** Multiple comparisons of Roll frequencies among wild-type (WT), *emx2* loss of function (LOF), and *emx2* gain of function (GOF) hair cells (HCs).

**Source data 2.** Quantification of Rock and Roll movements in wild-type (WT), *emx2* loss of function (LOF), and *emx2* gain of function (GOF) hair cells (HCs).

**Figure supplement 1.** Localization of Emx2 expression in wild-type (WT), loss of function (LOF), and gain of function (GOF) neuromasts at 2.0 dpf.

**Figure 4—video 1.** Time-lapse video of the wild-type (WT) (*myo6b:actb1-Gfp*) hair cell (HC) pair shown in *Figure 4A*.
https://elifesciences.org/articles/60432#fig4video1

**Figure 4—video 2.** Time-lapse video of a hair cell (HC) pair in wild-type (WT) (*myo6b:actb1-Gfp*) neuromast.
https://elifesciences.org/articles/60432#fig4video2

**Figure 4—video 3.** Time-lapse video of the *emx2* loss of function (LOF);*myo6b:actb1-Gfp* hair cell (HC) pair shown in *Figure 4B*.
https://elifesciences.org/articles/60432#fig4video3

**Figure 4—video 4.** Time-lapse video of the *emx2* gain of function (GOF); *myo6b:actb1-Gfp* hair cell (HC) pair shown in *Figure 4C*.
https://elifesciences.org/articles/60432#fig4video4

## Position of the Emx2-positive HC correlates with whether HC pairs undergo Roll or No Roll

The Rock and Roll results suggest that Emx2 may play a role in HCs prior to hair bundle establishment. Thus, we generated an *emx2:Gfp* reporter fish and used Gfp to track *emx2* promoter activity during live-imaging of Rock and Roll. To generate this reporter, we knocked in *Gfp* to the endogenous *emx2* locus using CRISPR (*Hoshijima et al., 2016a*; *Hoshijima et al., 2016b*). We found that Gfp expression in our *emx2:Gfp* reporter fish was present in the brain, pharyngeal arches, and otic vesicles, tissues that normally express *emx2* (*Figure 5—figure supplement 1*). *Gfp* expression in mature HCs was also consistent with normal Emx2 expression (*Jiang et al., 2017*), only present in A→P HCs of A-P neuromasts and in D→V HCs of D-V neuromasts (*Figure 5A*). Although not all Emx2-positive cells are necessarily Gfp-positive in the *emx2:Gfp* larvae, we did not observe any HCs with Gfp expression in the P→A, Emx2-negative HCs, further validating that Gfp reports *emx2* activity in our reporter.

Live-imaging of the *emx2:Gfp* larvae revealed that Gfp signal was first detectable after precursor division and in only one HC within a nascent HC pair (*Figure 5B*). With regard to the onset of Gfp expression, two expression patterns were observed during Rock and Roll in A-P neuromasts. In HC pairs that rolled, the posterior-positioned HC became Gfp-positive as it rolled into the anterior position (*Figure 5B*, Roll, 170 min, arrow, n = 4). In HC pairs that only rocked but did not undergo Roll, Gfp invariably turned on in the anterior HC (*Figure 5B*, No Roll, 156 min, arrow, n = 6). These results suggest that *emx2* is asymmetrically activated in one of the daughter HCs before the Roll phase is completed. The localization of Gfp in nascent HCs also supports our genetic results that Emx2 regulates the Rock and Roll (*Figure 4*).

## Emx2 is not required for positional acquisition of sibling HCs

Only 50% of HC pairs undergo HC rearrangement and Emx2 immunoreactivity was reported to be present in either the anterior or posterior nascent HCs initially (*Kozak et al., 2020*). These results suggest that identities of nascent HCs may be stochastically generated and that HCs utilize the rearrangement process to acquire their proper position (*Jacobo et al., 2019*; *Erzberger et al., 2020*). Our live-imaging results showed that posterior-positioned HCs turned on the *emx2* reporter as they rolled into the anterior position. Furthermore, anterior HCs that did not roll invariably turn on the reporter. These results are consistent with the hypothesis that HCs acquire their positions via the Rock and Roll process. Since Rock and Roll was affected in *emx2* mutants, we asked whether *emx2* is required for positional acquisition of HCs by assessing Gfp expression in *emx2* LOF (*emx2^{Gfp/-}*) neuromasts. We analyzed larvae at two dpf, in which the neuromasts exhibit only three to four pairs of HCs that are longitudinally aligned and HC pairs, hair bundle orientations, and positions can be easily identified (*López-Schier and Hudspeth, 2006*). Among the 160 WT HCs analyzed, reporter Gfp expression was only found in anterior-positioned HCs and no expression was found in the posterior-positioned HCs of A-P neuromasts (*Figure 5A*, *Table 1*). Surprisingly, among the 70 HCs we examined in *emx2* LOF neuromasts, all but one of 32 Gfp-positive HCs (half of the total number of HCs analyzed) were normally located in the anterior location of the neuromast (*Figure 5C–E*, LOF, *Table 1*). These results indicate that Emx2 is not required for HCs to acquire their normal position in the neuromast.

We also assessed Gfp expression in *emx2* GOF neuromasts in *emx2^{Gfp/+}* background (*emx2* GOF;*emx2^{Gfp/+}*). In the first generation (F1) of *emx2* GOF; *emx2^{Gfp/+}* neuromasts, only 24% of the total number of HCs analyzed were Gfp positive rather than the normal 49% in WT. This result suggests that the ectopic Emx2 expression in *emx2* GOF;*emx2^{Gfp/+}* neuromasts may suppress the endogenous *emx2* promoter activity driving the reporter. However, Gfp-positive HCs were always correlated with Emx2 immunoreactivity. Among the Gfp-positive HCs, 27% (8/30) were found to be mislocated in the posterior (*Figure 5F–H*, GOF, asterisk, *Table 1*). These results suggest that ectopic expression of Emx2, and likely in Emx2-negative HCs, affected the two sibling HCs' ability to acquire their proper locations within the neuromast. The reduction in the number of Gfp-positive cells and their mis-localization in the posterior region of the neuromast were no longer evident in larvae from F3 generation. This is most likely due to the reduced ectopic Emx2 expression (based on reporter mCherry expression) in the F3 generation, resulting from reduction in copies of the transgene from chromosomal segregation during cell divisions (*Table 1*). Nevertheless, the unidirectional hair bundle

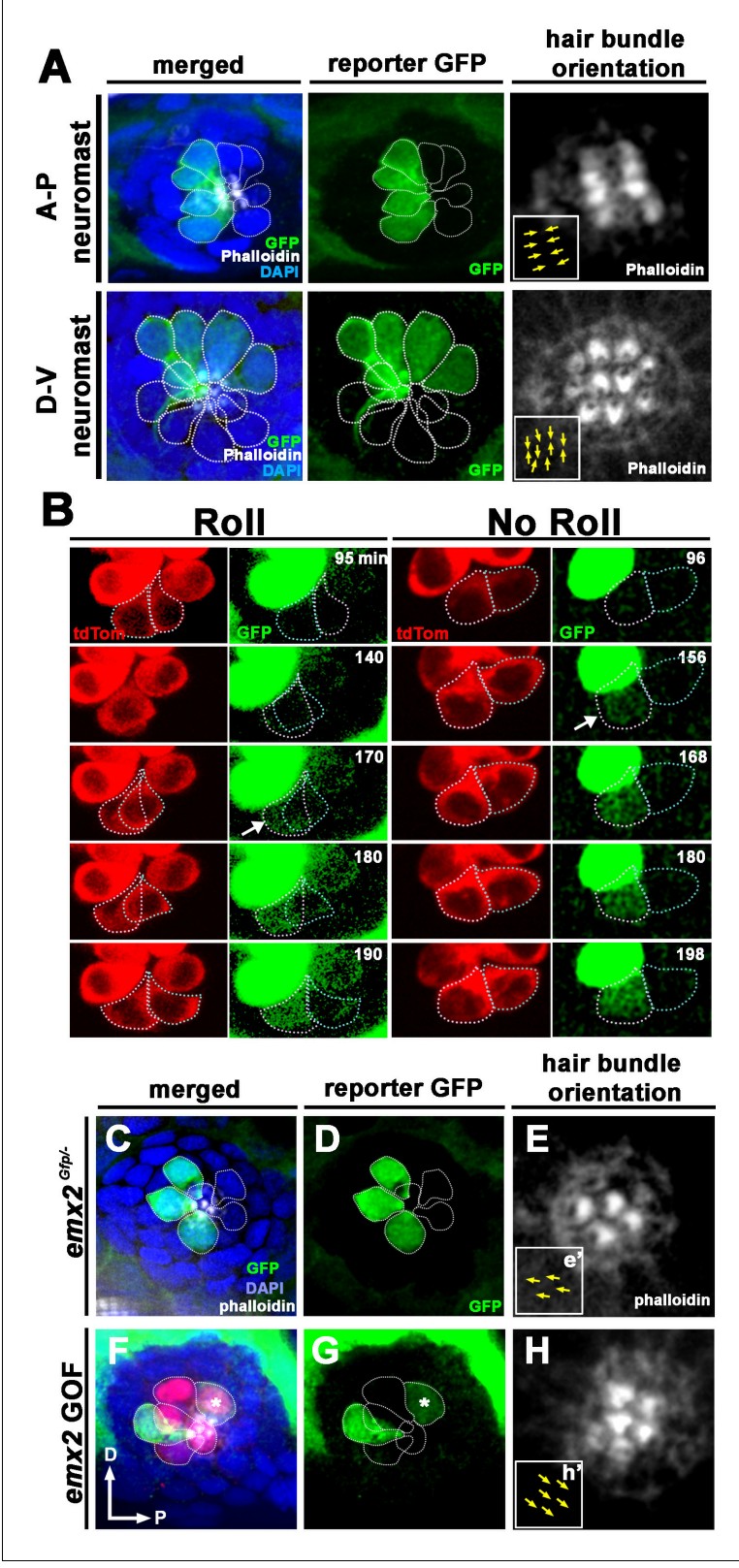

**Figure 5.** Spatiotemporal reporter activity of *emx2* in wild-type (WT) and *emx2* mutants. (**A**) In an *emx2:Gfp* zebrafish, Gfp-positive hair cells (HCs) are located in the anterior region of an A-P neuromast (top) and in the dorsal region of a D-V (bottom) neuromast. Phalloidin staining shows the hair bundle orientation (yellow arrows). (**B**) Time-lapse imaging of nascent HC pairs undergoing Roll or No Roll movement in *emx2:Gfp; myo6b:dtTomato*
*Figure 5 continued on next page*

*Figure 5 continued*

neuromasts. Left panel (Roll): A tdTomato-positive, nascent HC pair undergoing roll movement shows detectable Gfp expression in the posterior HC as it rolls into the anterior position (white arrow, 170 min into the Rock and Roll process). Right panel (No Roll): A HC pair that did not roll and Gfp appears in the anterior HC by 156 min into the Rock phase (white arrow). (C–E) The distribution of Gfp-positive HCs in an *emx2*$^{gfp/-}$ (loss of function [LOF]) neuromast, showing a merged image (C) of DAPI, GFP (D) and phalloidin staining (E). Outline of HCs are dotted. Gfp-positive HCs are located in the anterior region but hair bundles are pointing in P→A direction (yellow arrows in e'). (F–H) The distribution of Gfp-positive HCs in an *emx2* gain of function (GOF);*emx2*$^{Gfp/+}$ neuromast, showing a merged image (F) of DAPI, GFP (G), mCherry, and phalloidin staining (H). Among the total six pairs of GOF HCs analyzed, only two are Gfp-positive (G), one anterior- and one posterior (asterisk)-located. All hair bundles are in A→P direction (yellow arrows in h').

The online version of this article includes the following figure supplement(s) for figure 5:

**Figure supplement 1.** Generation of the *emx2:Gfp* transgenic line.

orientation phenotype remains largely similar between *emx2* GOF F1 and F3 larvae (data not shown).

## Emx2 affects HC morphology during Rock and Roll

Tracking *emx2* reporter activity in *emx2* LOF neuromasts indicates that positional acquisition of nascent HCs does not require Emx2 (*Figure 5C–E*). Rock and Roll, which is implicated in HCs' positional acquisition, was nevertheless affected in both *emx2* LOF and GOF nascent HCs (*Figure 4*). Then, what is the function of Emx2 and how does it regulate the Rock and Roll process? Previously, it was described that HCs form a 'bottle' shape prior to cell rearrangement (*Mirkovic et al., 2012*). It is possible that Emx2 is involved in regulating HC morphology, which then indirectly affects the Rock and Roll. To investigate this possibility, we used *myo6b:Yfp-tubulin* transgenic line to better visualize morphological changes in sibling HCs during Rock and Roll. Live-imaging revealed that after a HC precursor divided to form two daughter HCs, they were both initially spherical in shape (*Figure 6Bb*, 6 min). During the Rock phase, HCs started to extend a protrusion toward the apex next to the apical surface of mature HCs (*Figure 6Cc* and Dd, 18–33 min, arrow, *Figure 6—video 1*). It took approximately 30–40 min for the two nascent HCs to acquire the bottle-shape morphology after cell division, which was within the Rock phase. When the two nascent HCs rolled to exchange positions, the tip of the protrusion appeared to pivot at the apical surface of the neuromast (*Figure 6A–D*, Ee–Hh, *Figure 6—video 1*). These results suggest that apical protrusions may be important for efficient Roll movement.

Although maximum intensity projection (MIP) of WT live-images showed only moderate extension from the apex of nascent HCs in approximately 33 min (*Figure 6I–K*, arrowhead), three-dimensional (3-D) surface rendering of these MIP images revealed clearly that the protrusion was prominent and reached the apical surface of the neuromast within this time (*Figure 6i–k*, arrow). By contrast, *emx2* LOF images and 3-D rendering showed that nascent HCs took on a bottle-shape morphology by 6 min after cell division and the protrusion was wider and more pronounced than in WT (*Figure 6* Ll–Nn, arrowhead). Quantification of the tubulin-Yfp signals in the apical protrusion of WT and *emx2*

**Table 1.** Quantification of *emx2:Gfp* positive cells in wild-type (WT) and *emx2* mutant neuromasts.

| Genotype | Fish | Neuromast | Total hair cells (HCs) | Gfp+ cells | Anterior Gfp+ HCs | Posterior Gfp+ HCs | Mislocated Gfp+ HCs |
|---|---|---|---|---|---|---|---|
| WT | 7 | 20 | 160 | 78 (49%) | 78/78* (100%) | 0/78 (0%) | 0/78 (0%) |
| *emx2*$^{Gfp/-}$(loss of function [LOF]) | 3 | 10 | 70 | 33 (47%) | 32/33 (97%) | 1/33 (3%) | 1/33 (3%) |
| *emx2*GOF(F1); *emx2*$^{Gfp/+}$ | 10 | 18 | 123 | 30 (24%) | 22/30* (73%) | 8/30 (27%) | 8/30 (27%) |
| *emx2*GOF(F3); *emx2*$^{Gfp/+}$ | 29 | 29 | 246 | 100 (41%) | 98/100* (98%) | 2/100 (2%) | 2/100 (2%) |

*All Gfp+ cells are also positive for Emx2 immunostaining.

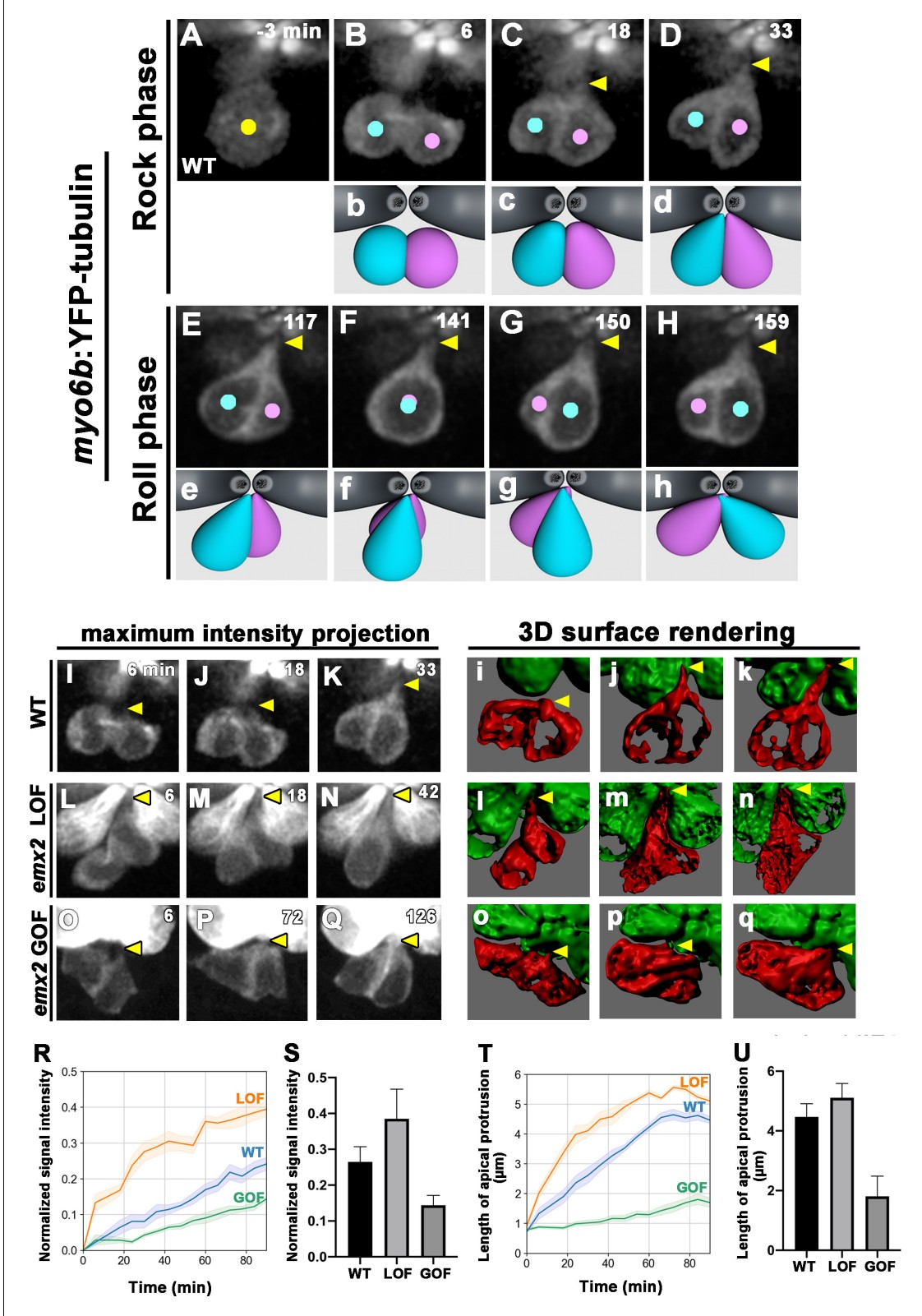

**Figure 6.** Emx2 affects apical protrusion formation in nascent hair cells (HCs). (A–H) Time-lapse images of *myo6b:YFP-tubulin* (wild-type [WT]) transgenic line showing nascent HCs undergoing Rock and Roll. After precursor cell divides (**A**), sibling HCs form an apical protrusion within 30 min of the Rock phase (**B–D**, yellow arrowhead, n = 6). When they exchange positions at the Roll phase, they appear to be pivoted at the apex (**E–H**, yellow arrowheads). Animation of images is shown in respective panels (Rock phase: **b–d**, Roll phase: **e–h**). (**I–Q**) Maximum intensity projection images

*Figure 6 continued on next page*

*Figure 6 continued*

generated by selected slices along z-axis in WT (**I–K**, same sample of **A–H**, n = 3), *emx2* loss of function [LOF] (**L–N**, n = 3) and *emx2* gain of function (GOF) (**O–Q**, n = 3). (**i–q**) Three-dimensional surface rendering of selected time points of live-imaging of HC pairs (red) in WT (**i–k**, n = 3), *emx2* LOF (**l–n**, n = 3), and *emx2* GOF (**o–q**, n = 3) neuromasts. Mature HCs are labeled in green. (**R**) Increases of YFP-tubulin signal intensity at the apical region of nascent HCs of WT (n = 5), *emx2* LOF (n = 5), and *emx2* GOF (n = 6) during the Rock phase (90 min). The shaded area represents the standard error of the mean (SEM). (**S**) YFP-tubulin signal intensities of the apical protrusion at the end of Rock phase were significantly different among WT (n = 5), *emx2* LOF (n = 5), and *emx2* GOF HCs (n = 6, one-way ANOVA, F = 26.97, p<0.0001; post-hoc Dunnett's multiple comparisons test for WT vs. LOF, p=0.0073, WT vs. GOF, p=0.0052. (**T**) Increases over time in the length of apical protrusion in WT (n = 5), *emx2* LOF (n = 5), and *emx2* GOF (n = 5) during the Rock phase (90 min). The shaded area represents the SEM. (**U**) The length of apical protrusion at the end of the Rock phase in WT (n = 5), *emx2* LOF (n = 5), and *emx2* GOF (n = 5) were compared. Significance was assessed by using one-way ANOVA, F = 52.15, p<0.0001. Post-hoc Dunnett's multiple comparisons test for WT vs. LOF is not significant (p=0.148) but significant for WT vs. GOF (p<0.0001). The following figure supplement, source data, videos are available for *Figure 6—figure supplement 1*. Measurements of signal intensity and length of the apical protrusion using Image J Fiji.

The online version of this article includes the following video, source data, and figure supplement(s) for figure 6:

**Source data 1.** Quantification of changes in signal intensity of apical protrusion of hair cells (HCs) during the Rock Phase.
**Source data 2.** Quantification of signal intensity of apical protrusion at the end of Rock phase among wild-type (WT), *emx2* loss of function (LOF), and *emx2* gain of function (GOF) hair cells (HCs).
**Source data 3.** Quantification of changes in the length of apical protrusion during the Rock Phase.
**Source data 4.** Quantification of the length of apical protrusion at the end of Rock phase among wild-type (WT), *emx2* loss of function (LOF), and *emx2* gain of function (GOF) hair cells (HCs).
**Figure supplement 1.** Measurements of signal intensity and length of the apical protrusion using ImageJ Fiji.
**Figure 6—video 1.** Time-lapse video of the *myo6b:yfp-tubulin* hair cell (HC) pair in *Figure 6A–H*.
https://elifesciences.org/articles/60432#fig6video1

LOF HCs using ImageJ Fiji plugin (*Figure 6—source data 1*) showed a stronger signal intensity in *emx2* LOF HCs during and by the end of the Rock phase than WT HCs (*Figure 6R and S*, *Figure 6— source data 2*). The extension of the apical protrusion in *emx2* LOF HCs also formed faster even though its length was similar to the WT HCs by the end of the Rock phase (*Figure 6T and U*). In contrast, in *emx2* GOF neuromasts, the two daughter HCs were initially spherical in shape similar to WT, but they took a much longer time to form the bottle shape and they only formed a small apical protrusion relative to the WT by 126 min (*Figure 6Oo–Qq*, arrowhead). Quantification of the tubulin-Yfp intensity in the apical protrusion of *emx2* GOF HCs also showed a slower rise in the Yfp signal and the length of protrusion, compared to the WT (*Figure 6R–U*, *Figure 6—source datas 1* and *2*).

Unfortunately, the live-images and 3-D rendering were unable to provide sufficient clarity to discern whether the apical protrusion originated in just one or both HCs in WT. Nevertheless, the pronounced apical protrusions in the LOF mutants and the stunted protrusions in the GOF mutants suggest that the Emx2-positive HC lacks apical protrusion during Rock and Roll and the protrusion in WT is primarily contributed by the Emx2-negative HC. Furthermore, the enhanced and reduced apical protrusions in the respective LOF and GOF mutants most likely affected the Rock and Roll process.

## Apical protrusion in nascent HCs is important for Rock and Roll

The apical protrusion of nascent HCs showed a high concentration of tubulin. We tested the importance of apical protrusion formation for the Rock and Roll by treating two dpf zebrafish larvae with nocodazole, which disrupts microtubule polymerization (*Hoebeke et al., 1976*). First, we investigated the effect of transient nocodazole treatments on the positioning of the two sibling HCs using *emx2:Gfp*. Under live-imaging, HC precursor divisions in neuromasts of *myo6b:Yfp-tubulin; emx2: Gfp* larvae were monitored and nocodazole was added immediately after a precursor cell divided to form two sibling HCs and treated for 90 min, which encompassed the duration of a normal Rock phase. During this treatment, sibling HCs often aligned along the D-V axis with each other (*Figure 7A*, 78–90 min, *Figure 7—video 1*), which were not observed in controls (*Figure 6*). After nocodazole removal, the sibling HCs resumed the Roll phase (*Figure 7A*, 96–174 min) and acquired their normal positions based on the Gfp-positive HC location in the anterior position by the end of 282 min (*Figure 7A*, arrowhead, n = 5, three Roll and two No Roll samples, *Figure 7—video 1*).

In contrast, extended treatments with nocodazole up to 150 min (*Figure 7B*, *Figure 7—video 2*) which encompassed the entire Rock and Roll duration normally did not result in positional exchange

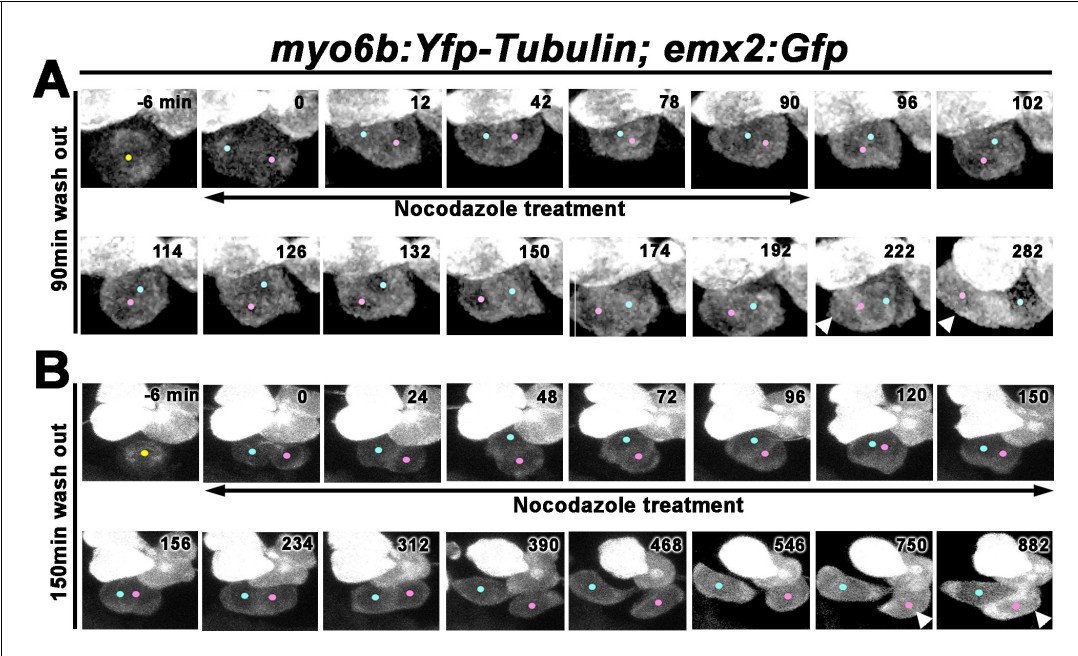

**Figure 7.** Nocodazole disrupts Rock and Roll and positional acquisition of hair cells (HCs). (**A**) Nascent sibling HCs in *myo6b:Yfp-tubulin; emx2:Gfp* larvae were treated with nocodazole for 90 min immediately after the HC precursor was observed to divide under live-imaging (0 min). After nocodazole removal, the nascent HC pair started to roll and they exchanged positions (96–174 min). The HC located at the anterior (pink dot) showed stronger Gfp signal due to the presence of *emx2:Gfp* and *Yfp-tubulin* alleles (222–282 min, arrowheads) than the *emx2:Gfp*-negative HC at the posterior (blue dot), which only expresses *Yfp-tubulin*. (**B**) Nascent HCs were treated with nocodazole for 150 min after HC precursor divided (0–150 min). After nocodazole removal, HCs failed to exchange their positions (156–312 min), which resulted in the mislocation of the *emx2:Gfp*-positive HC to the posterior (arrowheads, 618–882 min, pink dots).

The online version of this article includes the following video(s) for figure 7:

**Figure 7—video 1.** Time-lapse video of nocodazole treatment for 90 min of the *myo6b:Yfp-Tubulin; emx2:Gfp* hair cell (HC) pair shown in *Figure 7A*.
https://elifesciences.org/articles/60432#fig7video1

**Figure 7—video 2.** Time-lapse video of nocodazole treatment for 150 min of the *myo6b:Yfp-Tubulin; emx2:Gfp* hair cell (HC) pair shown in *Figure 7B*.
https://elifesciences.org/articles/60432#fig7video2

between sibling HCs after drug removal (*Figure 7B*, 156–882 min). Consequently, sibling HCs were often mispositioned with the *emx2:Gfp* HC located in the posterior (*Figures 7B*, 882 min, arrowhead, n = 4 mislocated *emx2:Gfp* HCs out of six samples analyzed).

Next, we investigated whether the recovery of the Roll phase was correlated with apical protrusion formation using the similar 3-D rendering approach as described for WT (*Figure 6I–q*). The 3-D rendering analyses were conducted using only *myo6b:Yfp-tubulin* larvae since the presence of the *emx2:Gfp* allele generated some background, which hindered reliable analyses. Using *myo6b:Yfp-tubulin* larvae, we observed that despite the nocodazole treatment starting immediately after cell division (*Figures 8A*, 0 min), apical protrusion was evident between 12 and 18 min into the treatment (*Figures 8A*, 18 min, red and white arrowheads, *Figure 8—video 1*), similar to untreated controls (*Figure 6J,j*). This protrusion disappeared by 30–36 min later (*Figures 8A*, 48 min, asterisk) but reappeared by 12 min after nocodazole was removed at 90 min (*Figures 8A*, 102 min, red and white arrowheads, n = 3, 2 Roll and one No Roll samples). Then, the two sibling HCs started to roll and exchange their positions by approximately 80 min after the apical protrusion was evident (*Figures 8A*, 150–186 min). In samples that were treated with nocodazole for 150 min, the apical protrusion was also apparent by 12–18 min of treatment (*Figures 8B*, 12 min, red and white arrowheads), which disappeared after 30–36 min (*Figures 8B*, 54 min), similar to HCs treated for 90 min (*Figure 8A*). However, no apical protrusion was observed after nocodazole removal at the end of 150 min, and sibling HCs failed to roll after 90 min (*Figures 8B*, 240 min, n = 5). These results are

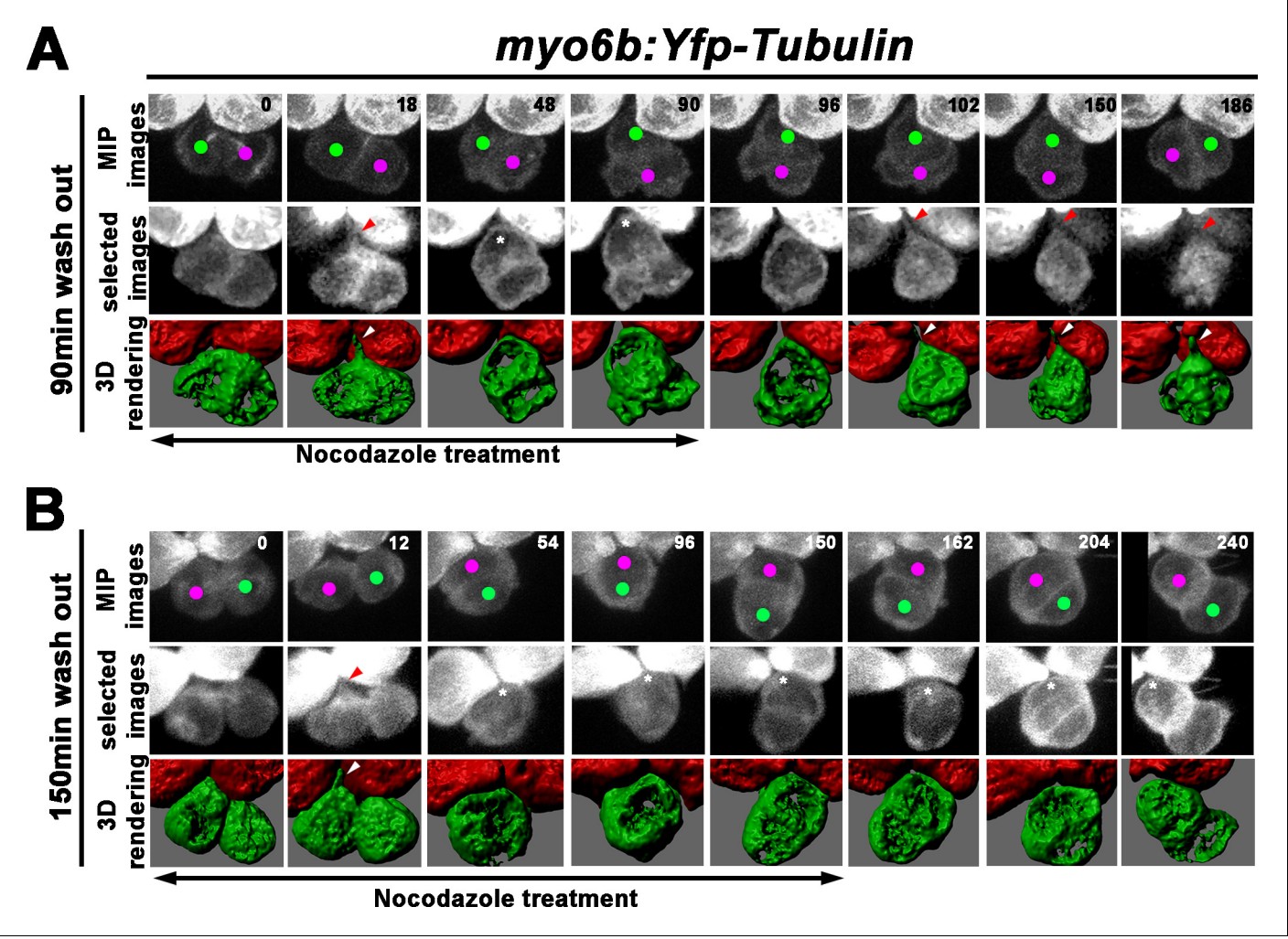

**Figure 8.** Formation of the apical protrusion is correlated with the Roll movement and hair cell (HC) positioning. (**A**) Maximum intensity projection and 3-D rendering of selected images of nascent sibling HCs in *myo6b:Yfp-tubulin* larvae (magenta and green dots) that was treated with nocodazole for 90 min immediately after the precursor divided. The apical protrusion that normally forms within 18 min was still evident initially (18 min; red and white arrowheads) but disappeared quickly between 30 and 36 min (asterisks, 48 min). After nocodazole removal, the apical protrusion reappeared after 12 min (102 min, red and white arrowheads) and the two nascent HCs rolled to their respective positions after 84 min (186 min). (**B**) A nascent HC pair (red and green dots) that was treated with nocodazole for 150 min immediately after the precursor divided. Similar to 90 min treatments, the apical protrusion was evident between 12 and 18 min (12 min, red and white arrowheads) after HC precursor divided (0 min) but disappeared shortly within 30–36 min (54 min, asterisk). However, after nocodazole removal at 150 min, sibling HCs did not form an apical protrusion and exchange positions (162–240 min, asterisks).

The online version of this article includes the following video(s) for figure 8:

**Figure 8—video 1.** Time-lapse video of nocodazole treatment for 90 min of a *myo6b:Yfp-Tubulin* hair cell (HC) pair shown in *Figure 8A*.
https://elifesciences.org/articles/60432#fig8video1

**Figure 8—video 2.** Time-lapse video of nocodazole treatment for 150 min of a *myo6b:Yfp-Tubulin* hair cell (HC) pair shown in *Figure 8B*.
https://elifesciences.org/articles/60432#fig8video2

consistent with the hypothesis that proper apical protrusion of nascent HCs is important for the Rock and Roll process and positional acquisition of HCs.

## cPCP pathway is important for Rock and Roll

Our nocodazole results suggest that the apical protrusion in nascent HCs is important for Rock and Roll. We further investigated the relationship between apical protrusion formation and the Rock and Roll process in mutants that are known to show abnormal hair bundle orientation in the neuromast

such as the *trilobite*, in which the cPCP gene *van gogh like 2* (*vangl2*) is mutated. cPCP pathway is a conserved intercellular mechanism to organize cells across the plane of a tissue (*Goodrich and Strutt, 2011*; *Wallingford, 2012*). Hair bundles in neuromasts of *trilobite* are misaligned and the Rock and Roll process is abnormal (*Mirkovic et al., 2012*). However, it is not known whether the abnormal Rock and Roll directly causes mispositioning of Emx2-positive and -negative HCs within the neuromast. We addressed this question by conducting time-lapse recording of the *trilobite*, *vangl2^{m209}*, in a *myo6b:β-actin-Gfp; emx2^{gfp/+}* background. Expression of Emx2 is not affected in *vangl2^{m209}* mutants and Emx2 and cPCP pathway are independently regulated (*Ji et al., 2018*). Thus, the GFP signal driven under the *emx2* promoter can be used to determine nascent HC positioning. Our results support previous findings that the frequency of Rock and Roll was decreased in the *vangl2^{m209}; myo6b:β-actin-Gfp; emx2^{gfp/+}* mutants, compared to controls (*Figure 9A*;

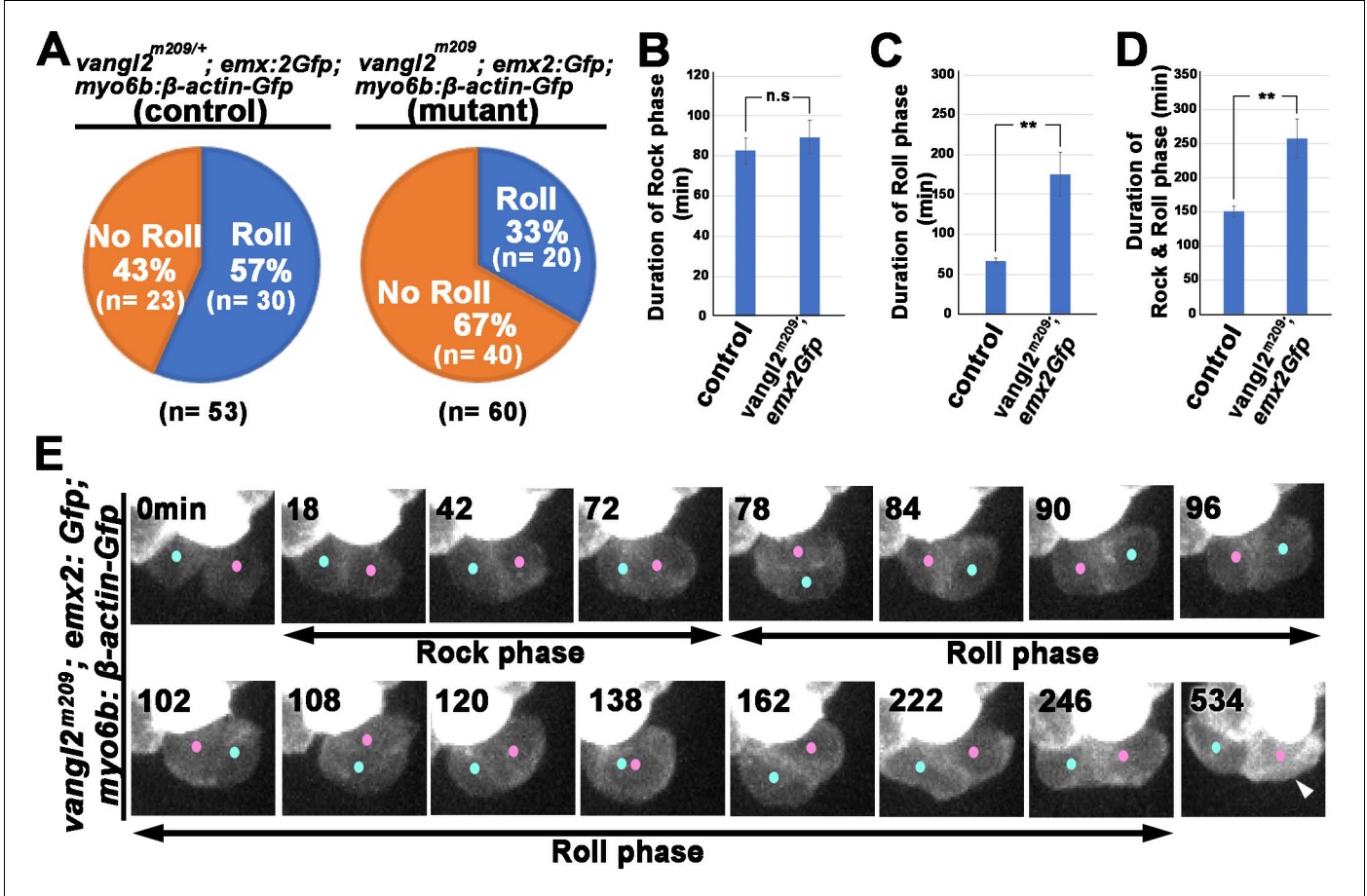

**Figure 9.** Rock and Roll is affected in the core planar cell polarity (cPCP) mutant, *trilobite*. (**A**) Frequencies of Roll and No Roll of nascent hair cell (HC) pairs in control (*vangl2^{m209/+}; emx2^{gfp/+}*) and mutant (*vangl2^{m209/m209}*; emx2^{gfp/+}) larvae. Significance was assessed by using chi-squared test with a 2 × 2 contingency table ($X^2$[(df = 1)]=6.1774, p<0.013, source data 1). (**B–D**) Duration of Rock (**B**), Roll (**C**), and Rock and Roll (**D**) of sibling HCs that underwent Rock and Roll in control (n = 28 from seven larvae) and *vangl2^{m209/m209}; emx2^{gfp/+}* (n = 19 from seven larvae) larvae. Significance was assessed by using Student's t-test (*p<0.05, **p<0.001, source data 2). The error bars represent SEM. (**E**) Time-lapse images of a HC pair in *trilobite* (*vangl2^{m209/m209}*); emx2^{gfp/+} mutant, which undergoes a prolong Roll phase from 78 to 246 min with several rolls (first roll: 78–96 min, second roll: 102–120 min, third partial roll: 138–162 min) and resulted in the *emx2:Gfp*-positive HC located to the posterior. The following figure supplement, source data, and video are available for *Figure 9—figure supplement 1*. Formation of apical protrusion in nascent HCs of *trilobite*.

The online version of this article includes the following video, source data, and figure supplement(s) for figure 9:

**Source data 1.** Frequencies of Roll and No Roll movements in *vangl2* mutants.

**Source data 2.** Comparison of duration of Rock, Roll, and R and R between controls and *vangl2* mutants.

**Figure supplement 1.** Formation of apical protrusion in nascent hair cells (HCs) of *trilobite*.

**Figure 9—video 1.** Time-lapse video of a *vangl2^{m209}; emx2: Gfp; myo6b: β-actin-Gfp* hair cell (HC) pair during Rock and Roll in *Figure 9E*.

https://elifesciences.org/articles/60432#fig9video1

*Mirkovic et al., 2012*). In HC pairs that underwent Rock and Roll, the duration of the Roll phase was prolonged (*Figure 9C and D*), even though the Rock phase was largely unaffected (*Figure 9B*). *Figure 9E* illustrates an example of a HC pair in *vangl2^{m209}* neuromast that underwent a prolonged roll phase and the Emx2-positive HC, as indicated by the additional Gfp signals driven by the *emx2* promoter, was mislocated to the posterior at the end of Roll phase (*Figure 9—video 1*). This mislocation of the Emx2-positive HCs was observed for HC pairs that did not undergo roll as well (*Table 2*). Furthermore, although the genetic background of *myo6b:β-actin-Gfp* is not ideal for visualizing the apical protrusion, we did not observe an obvious difference in the timing and formation of the apical protrusion in nascent HCs of *vangl2^{m209}*, compared to controls (*Figure 9—figure supplement 1*). Together, these results provided additional evidence that abnormal HC rearrangement is correlated with mispositioned HCs in the neuromast and that both intercellular signaling and apical protrusion formation are important for nascent HCs to undergo normal Rock and Roll.

## Discussion

In this study, we set out to determine whether Emx2 mediates the bidirectional HC pattern in the zebrafish neuromast (*Jiang et al., 2017*) by changing the location of its HCs within the neuromast via cell rearrangement or by changing the position of where hair bundle is established within its HCs, or both. First, based on our scRNA-seq analysis and live-imaging of *emx2:Gfp* reporter activity in control fish, we pinned *emx2* expression to the nascent HC stage well ahead of hair bundle establishment. These expression results support the notion that Emx2 is involved in positional acquisition of HCs. Additionally, the HC rearrangement process, Rock and Roll, was affected in both *emx2* loss- and gain-of-function mutants, thus providing the phenotypic evidence that supports the early onset of *emx2* expression (*Figure 1C*). However, our live-imaging results of *emx2:Gfp* HCs in *emx2^{Gfp/-}* neuromasts unequivocally demonstrated that the lack of Emx2 did not affect HCs' ability to acquire their respective positions (*Figure 5*). Since Emx2 does not confer positional information to HCs in the neuromast, we inferred that Emx2 mediates the bidirectional hair bundle pattern by changing the position where the hair bundle should be established within the HC. This effect of Emx2 on hair bundle establishment in neuromast HCs is most likely similar to utricular HCs in the mouse inner ear, which requires the heterotrimeric G proteins (*Jiang et al., 2017*).

The fact that Emx2 is not required for positional acquisition of HCs raised the question of how Emx2 regulates Rock and Roll. Our results showed that Emx2 regulates Rock and Roll by changing the shape of its nascent HCs, namely by reducing or delaying the apical protrusion formation of its nascent HCs. This delay in protrusion formation was only transient since all mature HCs ultimately formed apical protrusions at the vertex of the neuromast. In *emx2* LOF neuromasts, in which the apical protrusion was robust, HCs appeared to acquire their designated locations quicker (*Figure 4E*, shorter Rock and Roll duration). In contrast, in *emx2* GOF neuromasts, HCs lacked a clear protrusion and the duration of the Roll phase was longer than controls (*Figure 4F*) and sometimes HCs were mispositioned (*Table 1*). Extrapolating from the Rock and Roll and the 3-D rendering results, we postulate that the lack of apical protrusion from nascent HCs affected a stable attachment at the apical surface of the neuromast resulting in a longer duration of roll and occasional mispositioning of HCs. We further propose that in a normal neuromast, the apical protrusion of the Emx2-negative HC

**Table 2.** Summary of *emx2:Gfp*-positive hair cell (HC) positions in neuromasts after Rock and Roll.

| | Positions of *emx2^{Gfp}* HCs | | | | | | | |
| | Roll | | | | No Roll | | | |
| | Anterior | Posterior | Dorsal | Ventral | Anterior | Posterior | Dorsal | Ventral |
|---|---|---|---|---|---|---|---|---|
| Control* (%) | 30/30 (100%) | 0/30 (0%) | 0/30 (0%) | 0/30 (0%) | 19/19 (100%) | 0/19 (0%) | 0/19 (0%) | 0/19 (0%) |
| *vangl2^{m209}; emx2:Gfp; myo6:βactin-Gfp*** (%) | 5/17 (29.4%) | 6/17 (35.3%) | 4/17 (23.5%) | 2/17 (11.7%) | 13/39 (33.3%) | 13/39 (33.3%) | 7/39 (17.9%) | 6/39 (15.4%) |

*Four HC pairs with No Roll in **Figure 9A** did not express *emx2:Gfp* and were excluded from this table.

**Three HC pairs with Roll and one HC pair with No Roll in **Figure 9A** did not express *emx2:Gfp* and were excluded.

within a HC pair allows it to anchor at the vertex of the neuromast, whereas the non-anchored Emx2-positive HC is guided by the protruded HC for the positional exchange. Thus, the Emx2-negative HC is more important for the normal HC rearrangement process. We concluded that Emx2 functions to delay the apical protrusion of its HC, which contributes to the asymmetry between the two sibling HCs and facilitates the Rock and Roll process. Furthermore, Emx2's role in regulation of the Rock and Roll is independent of its role in hair bundle establishment since in F3 generation of *emx2* GOF neuromasts, HC positions are mostly normal but the bidirectional hair bundle-pattern is perturbed with most bundles pointing in the A→P orientation.

Our live-imaging results of *emx2:Gfp/+* reporter fish showing that the phenomenon of Roll or No Roll being consistent with the need or lack thereof to reposition the nascent Emx2-positive cell provided the direct evidence that HCs acquire their proper locations within the neuromast via the cell rearrangement process (*Figure 10*). Additionally, the hypothesis that apical protrusion formation in nascent HCs is important for Rock and Roll was further supported by the nocodazole results. The recovery of apical protrusion formation after nocodazole removal was correlated with the positional exchange of sibling HCs and correct positional acquisition, whereas failure to form an apical protrusion after drug removal resulted in the absence of roll movements and subsequent mispositioning of HCs (*Figures 7* and *8*).

What is the underlying mechanism for nascent HCs to acquire their positions within the neuromast since Emx2 is not involved? Accumulating evidence suggests that lateral inhibition via Notch signaling is important. In *notch1a⁻/⁻* mutants, all HCs are Emx2 positive with A→P hair bundle orientation (*Jacobo et al., 2019*; *Kozak et al., 2020*). Consistently, constitutive expression of the Notch intracellular domain results in neuromasts with more P→A HCs that are Emx2 negative (*Jacobo et al., 2019*; *Kozak et al., 2020*). More importantly, nascent HC pairs fail to undergo rearrangement in *notch1a⁻/⁻* mutants (*Erzberger et al., 2020*). Together, these results suggest that Notch signaling confers asymmetry and positional identity (PI) of sibling HCs, which mediates the Rock and Roll (*Figure 10*). This asymmetry includes differential regulation of *emx2*, upregulating *emx2* in the HC with anterior identity (Pla) and inhibiting/blocking *emx2* expression in the HC with posterior identity (Plp), which also contributes to the Rock and Roll.

Taking together the Notch results and our detection of high *emx2* expression at the nascent HC stage (Figure 3) suggest that the Notch signaling is functioning intercellularly between two sibling HCs. However, if Emx2 is indeed enriched in HC precursors as described (*Kozak et al., 2020*), then the mechanism of Notch signaling on regulating Emx2 would be different and will require the down-regulation of *emx2* in one of the sibling HCs (*Figure 1C*). Since expression of Emx2 in both daughter HCs was not observed with immunostaining (*Kozak et al., 2020*) or with *emx2:gfp* reporter, such

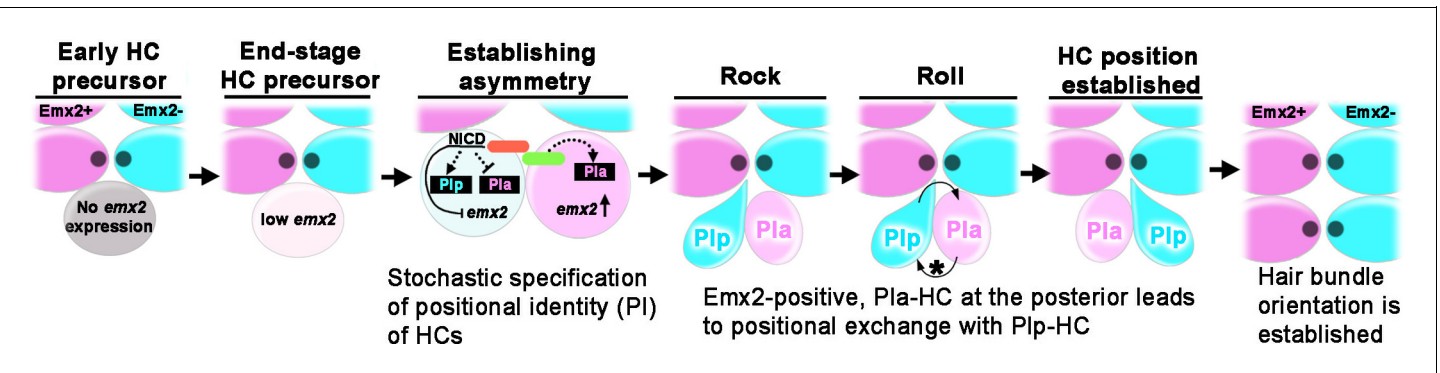

**Figure 10.** Model of positional acquisition of nascent hair cells (HCs) in the neuromast. Unlike scenarios 1 and 2 illustrated in *Figure 1C*, our results show a model in which an end-stage HC precursor before division expresses low levels of *emx2* (light pink). After the precursor divided, Notch signaling mediated through the Notch ligand (green color) and Notch1a receptor (red color) generates asymmetry between the two sibling HCs. This asymmetry results in stochastic specification of anterior (Pla) and posterior (Plp) positional identity (PI) to sibling HCs, which includes regulation of *emx2* expression. Thus, an *emx2*-positive, Pla-HC formed at the posterior position undergoes positional exchange with its sibling Plp HC, whereaswhen an *emx2*-positive, Pla-HC formed in the anterior position, the two sibling HCs do not exchange positions. In addition to the delay in apical protrusion mediated by Emx2, the Rock and Roll is also regulated by the intercellular core planar cell polarity (cPCP) pathway (asterisk).

downregulation may be quick and may involve asymmetric segregation of Notch determinants during precursor cell division (*Coumailleau et al., 2009*; *Kressmann et al., 2015*).

In addition to the Notch signaling, our results indicate that the cPCP pathway is also required for the Rock and Roll process. Emx2 and cPCP are independently regulated and *vangl2* is epistatic to *emx2* in hair bundle establishment (*Ji et al., 2018*). However, the cPCP pathway appears to function on multiple levels to generate the bidirectional HC pattern during neuromast development. For example, the cPCP pathway is important for HC precursor division since the axis of precursor division is affected in *vangl2^{m209}* mutants (*Mirkovic et al., 2012*). Second, HC pairs that did not undergo roll movements in *vangl2^{m209}* neuromast do show similar percentages of HCs that were mispositioned as HCs that rolled (*Table 2*). It is not clear if the mispositioned HCs in the No Roll pairs were caused by the failure of HCs to undergo the Roll phase or the cPCP's effects in the axis of precursor division or other functions indirectly affected both the Roll and No Roll HC pairs equally. Despite the possibility that the cPCP could function on multiple levels, the analyses of *vangl2^{m209}* mutants in *emx2:Gfp/+* reporter background indicates that abnormal Rock and Roll results in HC mispositioning.

What is the source of cPCP signaling for the HCs? Vangl2 appears to be expressed predominantly in HCs and not supporting cells (*Mirkovic et al., 2012*; *Jacobo et al., 2019*). Since the alignment of supporting cells is not affected in *vangl2^{m209}* neuromast (*Navajas Acedo et al., 2019*), it is likely that the intercellular signaling is occurring among sibling or neighboring HCs.

In summary, the sequential molecular events that give rise to a functional neuromast unit with bidirectional sensitivity to water flow is emerging. First, the Notch signaling imparts asymmetry between two nascent HCs, which brings the respective Emx2-positive and -negative HCs into position via the Rock and Roll process. While notch-mediated Emx2 is only one of the factors that facilitate this rearrangement process by creating asymmetry between the shape of the two sibling HCs, Emx2's major function is later, in establishing hair bundle orientation and neuronal selectivity (*Jiang et al., 2017*; *Ji et al., 2018*). Furthermore, cPCP proteins provide the intercellular signaling cues to maintain hair bundle alignment of HCs within the neuromast.

## Materials and methods

### Key resources table

| Reagent type (species) or resource | Designation | Source or reference | Identifiers | Additional information |
|---|---|---|---|---|
| Genetic reagent (*Danio rerio*) | Tg(myo6b:actb1-GFP) | PMC3426295 | RRID:ZFIN_ZDB-GENO-120926-20 | Kindt group (NIDCD, NIH) |
| Genetic reagent (*Danio rerio*) | Tg(myo6b:emx2-p2a-nls-mCherry) | PMC5388538 | RRID:ZFIN_ZDB-GENO-170619-3 | Wu group (NIDCD, NIH) |
| Genetic reagent (*Danio rerio*) | Tg(myo6b:YFP-Tubulin) | N.A | N.A. | Kindt group (NIDCD, NIH) |
| Genetic reagent (*Danio rerio*) | Tg(myo6b: tdTomato) | N.A | RRID:ZFIN_ZDB-TGCONSTRCT-160316-7 | Kindt group (NIDCD, NIH) |
| Genetic reagent (*Danio rerio*) | emx2 ko | PMC5388538 | RRID:ZFIN_ZDB-ALT-170606-5 | Kindt group (NIDCD, NIH) |
| Genetic reagent (*Danio rerio*) | Emx2:GFP | this paper | N.A. | Wu group (NIDCD, NIH) |
| Genetic reagent (*Danio rerio*) | Tg(myo6b: RiboTag) | PMC5939014 | PRID:ZFIN_ZDB-ALT-180129–10 | Kindt group (NIDCD, NIH) |
| Genetic reagent (*Danio rerio*) | Tg(myo6b:actb1-GFP) | PMC3426295 | RRID:ZFIN_ZDB-GENO-120926-20 | Kindt group (NIDCD, NIH) |
| Sequence-based reagent | tracrRNA | IDT | Cat# 1072533 | 25 µM |

*Continued on next page*

*Continued*

| Reagent type (species) or resource | Designation | Source or reference | Identifiers | Additional information |
|---|---|---|---|---|
| Sequence-based reagent | emx2_guide RNA 5'- atggaaagctgccgcgtcccagg-3' | IDT | this paper | 50 µM |
| Sequence-based reagent | tracrRNA | IDT | Cat# 1072533 | 25 µM |
| Recombinant DNA | pT3TS-nls-zCas9-nls | Addgene | Cat# 46757 | |
| Recombinant DNA | pKHR4 | PMC4806538 | | |
| Recombinant DNA | pKHR4-LHA-emx2-RHA | this paper | this paper | Wu group (NIDCD, NIH) |
| Antibodies | Rabbit anti-Emx2 | Trans Genic (fukuoka, japan) | KO609 | |
| Antibodies | Mouse anti-GFP | Thermo Fisher Scientific | A11120 | |
| Antibodies | Rabbit anti-GFP | Abcam | Ab6556 | |
| Antibodies | Alexa Fluor 647 phalloidin | Themo Fisher Scientific | A22287 | |
| Chemical compound, drug | Nocodazole | Sigma-Aldrich | M1404 | 10 µg/ml |
| Recombinant proteins | I-SceI | NEB | R0694S | |
| Software and algorithms | ImageJ | NIH | | https://imagej.net/Fiji/Downloads |
| Software and algorithms | ImageJ Fiji | NIH | | https://imagej.net/Fiji/Downloads |
| Software and algorithms | PoorMan3Dreg | | | http://sybil.ece.ucsb.edu/pages/software.html |
| Software and algorithms | MTrack J | *Meijering et al., 2012* | | https://imagescience.org/meijering/software/mtrackj/ |
| Software and algorithms | Fluorender 2.16 | University of Utah, SCI | | http://www.sci.utah.edu/software/fluorender.html |

## Zebrafish lines and genotyping

Adult zebrafish (*Danio rerio*) were maintained under a 14 hr light, 10 hr dark cycle. Zebrafish eggs and larvae were kept under the standard conditions at 28.5˚C in the E3 medium until desired developmental stages. Previously described transgenic zebrafish strains used in this study include the following: *Tg(myo6b:actb1-Gfp)^{vo8Tg}* (RRID:ZFIN_ZDB-GENO-120926-20), *Tg(myo6b:emx2-p2a-nls-mCherry)^{idc4Tg}* (RRID:ZFIN_ZDB-GENO-170619-3), *emx2^{idc5}(emx2 ko)* (RRID:ZFIN_ZDB-ALT-170606-5), *Tg(myo6b: tdTomato)* (RRID:ZFIN_ZDB-TGCONSTRCT-160316-7) (*Kindt et al., 2012*; *Jiang et al., 2017*), Tg(*myo6b: RiboTag*) (RRID:ZFIN_ZDB-ALT-180129-10) (*Matern et al., 2018*), Tg (*myo6b:YFP-Tubulin*) (gift from Katie Kindt), and *vangl2^{m209}* (RRID:ZFIN_ZDB-GENO-100615-1) (*Jessen et al., 2002*).

Emx2 LOF zebrafish were genotyped by PCR using the primers emx2-fPCR-Fwd (TCACTTAAAC TGGGGAATCTTGA) and emx2-fPCR-Rev (GGAGGAGGTACTGAATGGACTG). The PCR product was digested with restriction enzyme FauI, which cleaved the mutant PCR product into 200 bp and

100 bp and distinguished them from the WT product of 300 bp. Prior to live-imaging experiments, genotype of *emx2* LOF larvae was verified based on the position of the kinocilium that was illuminated by β-actin-GFP or Yfp-Tubulin signals under a fluorescent microscope. For *emx2* GOF larvae, the genotype was verified with mCherry expression in HCs as well as Gfp expression driven by cardiac myosin light chain (*cmlc*) promoter in the heart of the transgenic fish. The position of the kinocilium was verified based on β-actin-GFP or YFP-tubulin signal. The *vangl2^{m209}* allele was genotyped by PCR using the primers vangl2-PCR-Fwd (TAGGCCTGCATCTAACCAAAC) and vangl2-PCR-Rev (CCAGAAATGCCTGACCACAGATTC) as described (*Jessen et al., 2002*). The PCR product was digested with restriction enzyme AlwnI, which cleaves the mutant PCR product into 150 bp and 100 bp and distinguishes them from the WT product of 250 bp.

Generation of *Tg(emx2:Gfp)* fish *emx2:Gfp* transgenic fish (ZFIN ID: ZDB-GENE-990415–54) were generated using CRISPR and homologous recombination as described (*Hoshijima et al., 2016a*; *Hoshijima et al., 2016b*). Guide RNA (gRNA) target sites (5'-cctgggacgcggcagctttccat-3') were identified using the web program CHOPCHOP (http://chopchop.cbu.uib.no/index.php). The gRNA (emx2-gRNA: 5'- atggaaagctgccgcgtcccagg-3') identified by CHOPCHOP was used to design crRNA using IDT custom gRNA design website (https://www.idtdna.com/site/order/designtool/index/CRISPR_CUSTOM). Each custom designed crRNA (CRISPR RNA) and tracrRNA (trans-activating crRNA) was dissolved in duplex buffer (IDT) to a 50 µM stock solution and stored in −80℃. To prepare the crRNA:tracrRNA complex, equal volumes of 50 µM Alt-R crRNA and 50 µM Alt-R tracrRNA stock solutions were mixed together and annealed by heating at 70℃ for 5 min followed by gradual cooling at room temperature. The 25 µM crRNA:tracrRNA duplex stock solution was stored at −20℃. For *Cas9* mRNA, the zebrafish codon optimized *cas9* plasmid pT3TS-nls-zCas9-nls was used as template (*Jao et al., 2013*). The template DNA was linearized by XbaI and purified using a QIAprep purification column (Qiagen) and 500–1000 ng linearized template was used to synthesize capped RNA using the mMESSAGE mMACHINE T3 kit (Life Technologies) and precipitated using 5 M LiCl and diluted to 500 ng/µl.

The donor plasmid for targeted reporter gene integration was constructed by assembling donor sequences into pKHR4 vectors (provided by Dr. Hoshijima). eGFP was flanked with left and right homologous arms of approximately 1.0 kb of *emx2* genomic sequences upstream and downstream of the integration site, respectively (*Figure 3—figure supplement 1A*). Column-purified donor plasmid DNA was further purified to remove all traces of RNase activity by phenol/chloroform extraction, chloroform extraction, and ethanol precipitation with sodium acetate. Plasmid DNA was dissolved in nuclease-free water, quantified, diluted to 500 ng/µl, and stored at −20℃. Prior to injection into one-cell stage zebrafish embryos, 500 ng of donor plasmid DNA was digested with *I-SceI* enzyme. After enzyme digestion, a mixture of 5 µl of donor DNA solution (100 ng/µl) with 2.5 µl of Cas9 mRNA (500 ng/µl) and 1.5 µl of 25 µM crRNA:tracrRNA plus 1.0 µl of phenol red was kept on ice and 1 nl of the mixture was injected into the cytoplasm of zebrafish eggs at one-cell stage.

## Larvae dissociation and FACS

Zebrafish larvae *Tg(myo6b: RiboTag)* at four dpf were dissected to remove the head, which included the otic vesicles. Approximately 400 dissected bodies in batches of 50 were dissociated by adding 0.5 ml of 0.5% trypsin (Gibco, cat# 15400054) and triturated with 1 ml pipette tip for 30 s at 30℃. The dissociated cells were filtered with 40 µm strainer (Pluriselect, cat# 43-50040-50) and washed with PBS twice before adding DMEM containing 10% FBS and spun down by centrifugation at 700 × g for 5 min at 4℃. Cells were resuspended and kept in DMEM before processing for FACS, and cells were gated based on size and Gfp signal during FACS.

## 10× Genomics scRNA-seq library construction

scRNA-seq was carried out with 10× Genomics single cell platform (10× Genomics, Pleasanton, CA. USA). Approximately 5000 live cells after FACS in a maximum volume of 34 µl were loaded on a 10× Genomics Chromium Controller (10× Genomics). Chromium Single Cell 3' Library and Gel Bead Kit v2 (10× Genomics) was used for libraries preparation according to manufacturer's instructions. Quality of the libraries was evaluated on a Fragment Analyzer instrument (Agilent) and sequenced on a NextSeq 500 sequencer Illumina with the following paired read lengths: 26 bp Read 1, 8 bp I7 Index, and 98 bp Read 2.

### Single-cell RNA-seq read alignment and quantification

Raw reads were demultiplexed and aligned to version 10 of the zebrafish genome (GRCz10) using the Cell Ranger pipeline from 10× Genomics (version 3.0.1). A total of 4824 valid cell barcodes were obtained in ZM4 data set. The resulting barcodes (henceforth referred to as cells) were used to generate a UMI count matrix for further analyses. Data deposition: the BAM files and count matrices produced by Cell Ranger has been deposited to the Gene Expression Omnibus (GEO) database, http://www.ncbi.nlm.nih.gov/geo (accession no. GSE152859).

### Quality control, dimensional reduction, and cell clustering

Quality control, dimensional reduction, and cell clustering were performed using Scanpy (version 1.4.4) (*Wolf et al., 2018*). Standard quality control steps were used to filter noisy and unreliable cells. Notably, cells with the following descriptions were excluded from the pool: (1) less than 400 unique genes, (2) more than 2500 unique genes, (3) more than 10,000 UMI, and (4) more than 5% mitochondrial genes (indicator of cellular stress). Genes found in at least three cells were included in the analysis. After processing for quality control, 2882 cells were used for subsequent analyses. We followed the basic library size normalization to counts per million and the gene expression matrix was log-transformed before processing for principal component analysis (PCA). Following dimensional reduction, Louvain clustering was applied testing a range of vaules (0.2–2.0 in 0.2 increments) for the resolution. Resolutions of 0.4 and 0.5 were used for the results shown for the clustering and reclustering, respectively.

### Pseudotime analysis

Cells after reclustering were subjected to DPT analysis using Scanpy (version 1.4.4) (*Wolf et al., 2018*). Cluster #4 annotated as 'HC precursors' was selected as a root cluster to generate pseudo-time trajectory of HC differentiation. Expression levels of selected marker genes were represented by a heat map and normalized each gene to the value between 0–1.

### Live-imaging

Zebrafish larvae at 2.5–3.0 dpf were anesthetized with 0.03% ethyl 3-aminobenzoate methanesulfonate (Sigma, #MS-222, St Louis, MO) and mounted in a glass bottom dish (MatTek, P35G-0–14 C) with 1.0% low melting point agarose containing 0.01% tricaine. Approximately one to six neuromasts per animal were sequentially imaged with an automatic stage positioner. Neuromasts containing one to three pairs of mature HCs only were selected for time-lapse recording and samples were imaged using a 63X/NA1.4 oil-immersion objective lens on a spinning-disk confocal system (Ultraview, Perkin-Elmer, Waltham, MA) or a Zeiss LSM780 confocal microscope. Z-stacks were collected at intervals of 3–6 min. All images were processed with ImageJ Fiji (National Institutes of Health, Bethesda, MD, https://imagej.net/Fiji/Downloads).

### Analyses of nascent HCs during Rock and Roll

Analysis of nascent HCs during Rock and Roll was performed in ImageJ Fiji (*Schindelin et al., 2012*). After acquisition of z-stack images of the Rock and Roll process over time, rolling ball algorithm and Gaussian blur filter were applied to subtract the background signal from the images. PoorMan3Dreg (http://sybil.ece.ucsb.edu/pages/software.html) was used to correct artifacts for sample drift during the course of the recording. After sample drift correction, the center region for nascent HCs from the z-axis of optical sections was selected using the Substack Maker plugin, and the region of interest (ROI) was cropped using rectangular selection tool.

HC precursors and nascent HCs were tracked overtime of the HC rearrangement with ImageJ Fiji plugin MTrack J (https://imagescience.org/meijering/software/mtrackj/) (*Meijering et al., 2012*). Based on the cell tracking data of nascent HCs undergoing the rearrangement, we determined the duration of the Rock and Roll phases using the following criteria (*Figure 4A'*). The initiation of Rock phase was defined as the time immediately after the HC precursor divided. The line drawn between the center of the two HCs at the end of precursor cell division was considered as the horizontal axis, regardless of the position of the precursor within the neuromast. The angle of displacement of the two nascent HCs was within 30° in either direction of this horizontal axis during the Rock phase. When the angle displacement between the two HCs was greater than 30° and continued to increase,

it was considered as the beginning of the Roll phase and the end of the Rock phase. When the two HCs rotated 180˚ to switch positions and the cell–cell contact between the two HCs intersected with the horizontal axis, it was considered as the end of the Roll phase.

## Immunofluorescence and phalloidin staining

For immunostaining, 2.5–5.0 dpf zebrafish larvae were fixed with 4% paraformaldehyde in PBS for 3.5 hr at 4˚C. Post-fixed larvae were washed with 0.05% Tween-20 PBS (PBT) and treated with pre-chilled acetone at −20˚C for 2 min. Then, larvae were incubated with a blocking solution (2% goat serum, 1% BSA in PBT) for 2 hr at room temperature, followed by incubation with primary antibodies diluted in blocking solution for either 2 hr at room temperature or overnight at 4˚C. After incubation, larvae were washed four times with PBT for 5 min each before incubating with secondary antibodies at 1:1000 dilution in blocking solution for 2 hr at room temperature. Then, larvae were washed and mounted with Anti-fade and imaged on a Zeiss LSM780 confocal microscope using a 63X/NA1.4 oil-immersion objective lens. The following primary antibodies were used for immunostaining: rabbit anti-Emx2 (1:250; KO609, Trans Genic, Fukuoka, Japan), mouse anti-GFP (1:1000; A11120, Thermo Fisher Scientific, Carlsbad, CA), and Rabbit anti-GFP (1:1000; ab6556, Abcam). F-actin was detected with fluorescein or Alexa Fluor 647 phalloidin (1:50; Thermo Fisher Scientific, Carlsbad, CA).

## In situ hybridization

Whole mount in situ hybridization of zebrafish larvae was performed as described previously (*Thisse and Thisse, 2008*). In situ probes for *emx2* (IMAGE 7403786) were prepared as described (*Jiang et al., 2017*).

## 3-D surface rendering of nascent HCs

To acquire 3-D image data of apical protrusion formation in nascent HCs over time, 2.5 dpf zebrafish larvae of *Tg(myo6b:YFP-Tubulin)* were mounted in 1.0% low-melting point agarose containing 0.01% tricaine in a glass-bottom dish. Time-lapse imaging of apical protrusion formation was performed using 1 μm step-scan captured with 3 min interval for 2–3 hr or overnight. For 3-D surface rendering, specific time points were chosen from the acquired 4-D data sets and processed for background subtraction and noise reduction using ImageJ Fiji before performing threshold-based segmentation to distinguish nascent HCs from mature HCs using Fluorender 2.16 (University of Utah, SCI, http://www.sci.utah.edu/software/fluorender.html). Imaris 9.15 (Bitplane) was used to surface render three dimensionally the segmented images of nascent and mature HCs using the level of 0.15 in surface area detail.

## Quantification of Yfp-tubulin signal in the apical protrusion

MIP images from selected slices during the Rock phase were converted into 8-bit gray scale images using ImageJ Fiji (*Figure 6—figure supplement 1A*). Each image was converted into Rainbow RGB color mode from Lookup table of ImageJ Fiji for better visualization of the apical protrusion (*Figure 6—figure supplement 1B*). Based on the Yfp intensity threshold, both the ROI within the apical protrusion was automatically selected using the wand section tool and its mean intensity signal was obtained using the plug-in of ImageJ Fiji (*Figure 6—figure supplement 1C*). To measure the length of the protrusion, the selected region within the apical protrusion was converted into a mask image, in which a horizontal line was drawn across the top of the two HC nuclei (*Figure 6—figure supplement 1D*, blue line) and the length of the apical protrusion was measure by drawing a line from the tip of the protrusion perpendicular to the horizontal line (*Figure 6—figure supplement 1D*, pink line).

## Nocodazole treatments

A stock solution of nocodazole (Sigma, M1401-10MG) was prepared by dissolving 10 mg of nocodazole in 1.0 ml of DMSO, which was diluted to a working solution of 10 μg/ml in E3 medium. *Tg(myo6b:Yfp-tubulin, myo6b:Yfp-Tubulin;emx2:Gfp)* zebrafish larvae at 2.0 dpf were mounted in 1.0% low melting point agarose in a glass bottom dish and the neuromasts were live-imaged under a Zeiss 780 confocal microscope. Under time-lapse recording, when a HC precursor was observed to divide, 2 ml of nocodazole working solution was added to the dish immediately. Under continuous

recording, nocodazole was removed after either 90 min or 150 min of incubation with three washes of E3 medium and recording was terminated after the Rock and Roll process was completed or *emx2* reporter GFP expression was evident in one of the HCs within a HC pair.

## Quantification and statistical analysis

Statistical analyses of our quantification were performed using Prism 8, Microsoft excel, and R version 3.6.3. MANOVA with post-hoc Tukey's Honestly Significant Difference was used for multiple comparisons of duration of Rock and Roll in WT and *emx2* mutants. For frequency of Roll or No Roll of WT and *emx2* mutants, chi-square with $3 \times 3$ contingency table was applied and post-hoc chi-squared tests were performed for pairwise comparisons with FDR correction. Errors bars represent standard deviation (SD) in all graphs unless indicated otherwise.

## Acknowledgements

We thank Dr. Katie Kindt (NIDCD, NIH) for the gift of Tg(*myo6b: tdTomato),* Tg(*myo6b: RiboTag*), and Tg(*myo6b:YFP-Tubulin*) fish, Dr. Hoshijima for the pKHR4 plasmid, consultation for the generation of the *emx2: GFP* reporter zebrafish, Ms. Martha Kirby at NHGRI/DIR Flow Cytometry Core for conducting the FACS, Dr. Hui Cheng at NIDCD for statistics consultation, Genomics and Computational Biology Core (GCBC, NIDCD) for conducting $10\times$ Genomics experiments and sequencing. This work utilized the computational resources of the NIH HPC Biowulf cluster (http://hpc.nih.gov). We also thank Drs. Matthew Kelley, Katie Kindt, Robert Morell, as well as members of the Wu lab for their critical reading of the manuscript. This research was supported by the Intramural Program at NIDCD, NIH (Grants# 1ZIADC000021 to DKW and ZICDC000086 to GCBC).

## Additional information

### Competing interests

Doris K Wu: Reviewing editor, *eLife*. The other authors declare that no competing interests exist.

### Funding

| Funder | Grant reference number | Author |
| --- | --- | --- |
| National Institutes of Health | 1ZIADC000021 | Doris K Wu |
| National Institutes of Health | ZICDC000086 | Daniel Martin |

The funders had no role in study design, data collection and interpretation, or the decision to submit the work for publication.

### Author contributions

Sho Ohta, Conceptualization, Data curation, Software, Formal analysis, Validation, Investigation, Visualization, Writing - original draft, Writing - review and editing; Young Rae Ji, Conceptualization, Data curation, Formal analysis, Validation, Visualization, Writing - original draft, Writing - review and editing; Daniel Martin, Data curation, Formal analysis, Validation, Investigation, Visualization, Writing - review and editing; Doris K Wu, Conceptualization, Resources, Supervision, Methodology, Writing - review and editing

### Author ORCIDs

Sho Ohta ⓘ https://orcid.org/0000-0002-2421-8093
Young Rae Ji ⓘ https://orcid.org/0000-0002-8825-9783
Daniel Martin ⓘ https://orcid.org/0000-0002-8880-9087
Doris K Wu ⓘ https://orcid.org/0000-0002-1400-3558

### Ethics

Animal experimentation: All zebrafish experiments were conducted according to NIH approved animal protocol (#1362-13) and NIH animal user guidelines.

Decision letter and Author response
Decision letter https://doi.org/10.7554/eLife.60432.sa1
Author response https://doi.org/10.7554/eLife.60432.sa2

## Additional files

### Supplementary files
- Transparent reporting form

### Data availability
Sequencing data have been deposited in GEO under accession code GSE152859.

The following dataset was generated:

| Author(s) | Year | Dataset title | Dataset URL | Database and Identifier |
|---|---|---|---|---|
| Ohta S, Ji YR, Martin D, Wu DK | 2020 | Emx2 defines bidirectional polarity of neuromasts by changing hairbundle orientation and not hair-cell positions | https://www.ncbi.nlm.nih.gov/geo/query/acc.cgi?acc=GSE152859 | NCBI Gene Expression Omnibus, GSE152859 |

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
