## [Decision Letter]

**Acceptance summary:**

The work explores the role of Emx2 transcription factor in neuromast development. Through functional experiments and life imaging of Emx2 expression during the cell rearrangement process, this study shows that Emx2 is not required for cell positioning but rather for hair cell bundle orientation. The work nicely uses a new reporter line of Emx2 for life imaging. In addition, the study contributes to understanding of how cell shape is coupled to the cell arrangement process and expands the knowledge of Emx2 function by showing that Notch and the PCP pathway regulate cell rearrangement independent of Emx2.

**Decision letter after peer review:**

Thank you for submitting your article "Emx2 regulates hair cell rearrangement but not positional identity within neuromasts" for consideration by *eLife*. Your article has been reviewed by three peer reviewers, and the evaluation has been overseen by Marianne Bronner as the Senior Editor. The following individual involved in review of your submission has agreed to reveal their identity: David W Raible (Reviewer #3).

The reviewers have discussed the reviews with one another and the Reviewing Editor has drafted this decision to help you prepare a revised submission.

Summary:

This manuscript by Ohta and colleagues describes the development of lateral line hair cells and the cellular rearrangements that organize hair cells that will eventually extend oppositely oriented stereociliary bundles so that they are aptly positioned on the anterior or the posterior side of the neuromast. Since the transcription factor Emx2 regulates the orientation of the polarized stereociliary bundle it is hypothesized that Emx2 may similarly regulate hair cell rearrangement. The authors find that Emx2 regulates the development of aspects of hair cell morphology that influence rearrangements but does not appear to regulate the process directly. While all three reviewers found the topic of interest, they all agreed that more work was necessary to convincingly make the points. The necessary revisions are nicely described in the individual reviews, which we hope will be helpful in revising the paper.

Reviewer #1:

The manuscript by Ohta and colleagues deals with the role of the transcription factor Emx2 in organizing the bidirectional polarity of hair cells of the zebrafish neuromasts. Previous reports by the authors have identified Emx2 as responsible of the establishment of stereociliary bundle orientation in the vertebrate (mouse, chick, zebrafish) inner ear and zebrafish neuromasts. A recent report from Lopez-Schier lab suggests that the cell rearrangement process of nascent hair cell sibling cells to position them in the neuromast might be coupled with differential Emx2 expression to ensure their correct position within the neuromast. In this manuscript, the authors try to follow this question of whether Emx2 is required for sibling cell rearrangement and thus cell positioning or, alternatively, regulates bundle orientation as it does in the inner ear. For this, the authors generate a zebrafish reporter line of Emx2 to follow the expression of Emx2 during the rotation process and they also analyse cell rotation in Emx2 Lof and Gof embryos through life imaging. The authors conclude that while the rotation process is modified in Lof and Gof embryos (duration, phases), Emx2 does not impair the rotation process and thus not determine the position of sibling cells. Live imaging also allowed the authors to see differences in apical protrusions of Emx2-positive and Emx2-negative cells.

The manuscript is well performed and digs into an interesting question but fails short in their analysis of Emx2 function and whether the rotation of siblings has any meaningful role.

In general, the manuscript could improve the Introduction and Discussion text to discuss the different scenarios for *emx2* role, role of the rotation, apical protrusion, downstream players of *emx2* for bundle orientation.

1) By live imaging, the authors describe the % of cells performing cell rearrangement ( 57%) and the percentage that do not (43%) and separate this process into a rock phase and a roll phase (inversion). The authors find that when Emx2 is not expressed (in -/-) or overexpressed, there are differences in the time spent in each phase but cell rearrangement still happens. One would assume that if *emx2* differential expression makes cell´s to rearrange in order to segregate in space the ones positive Emx2 and the ones negative for Emx2 (as observed using the reporter line), no rearrangement would happen if the siblings are equal.

The authors find that in Lof and Gof embryos, cell rearrangement takes place. As a control it is necessary to show that in these conditions all cells are either positive or negative for Emx2. If still minor differences of Emx2 expression between siblings exist, cell rearrangement might not be blocked for this reason. Is Emx2 overexpression clearly affecting all cells, and Lof? It would be good to have a picture showing Emx2 immunoreactivity or ISH in *emx2*^-/-^ neuromasts and in Gof embryos.

In their previous published work (Jiang et al., 2017) they show Emx2 expression in Lof and Gof but at 4 dpf. In this case, I understand that the embryos are younger, at 2-3 dpf. Could the myo6 promoter driving Emx2 for Gof be milder at these stages?

Most probably the conclusion that Emx2 does not directly regulate cell rearrangement is correct but this additional control would improve the manuscript.

Another recent paper (Erzberger et al., 2020) has shown that cell rearrangement is impaired when siblings do not have differential activity of Notch pathway. Thus, being Emx2 downstream of Notch, the present results suggest that cell rearrangement mediated but Notch is driven by other downstream factors. The fact that probably Emx2 is not required for cell rearrangement but another factor is interesting.

2) The analysis with the reporter indicates that GFP appears when cells perform cell rearrangement and is always correlated with the final position cells adopts, if *emx2* positive are in posterior perform cell rearrangement, if are anterior located do not. In Gof, GFP cells can be found mislocated. This is a very nice result.

3) How the authors envisage that Emx2 regulates apical protrusions and cell shape changes? Can the authors affect apical protrusion formation to see if cell rotation is impaired?

4) Previously, the authors indicated that in zebrafish utricule and cristae, *emx2* is also involved in hair bundle polarity. I wonder if there a cell rearrangement between siblings also takes place as in the neuromasts. Live imaging in zebrafish inner ear could help to assess whether *emx2* role in the rock and roll process and cell shape is specific of neuromasts organization or has a wider role. Several manuscripts have been published in relation to the cell rearrangement process in the neuromasts and the mechanisms behind it, however, its biological implications if is very particular of this system, might be of low relevance. I think it is more relevant to explore deeply how *emx2* changes bundle orientation.

5) The authors perform scRNA-seq to trace the *emx2* expressing cells and establish the early expression of this transcription factor. Their analysis does not differ from the scRNAseq analysis and trajectory plot recently published in Kozak et al. and thus as it is presently, it does not add new information. I suggest eliminating this part or, if kept, use it to expand the analysis and perform functional experiments with putative downstream factors present in *emx2* expressing cells that could regulate sterociliary bundle orientation. A deeper study of the implications of *emx2* activity in bundle reorientation would make this manuscript more solid.

Reviewer #2:

Here, the authors find that Emx2 regulates the development of aspects of hair cell morphology that influence rearrangements but does not appear to regulate the process directly. The work is largely descriptive and does not reveal a mechanistic or transcriptional link between Emx2 and hair cell morphology nor does it consider the contribution of other signaling pathways such as PCP signaling which also controls stereociliary bundle orientation in the neuromast. For these reasons it seems an minor advance and is currently not comparable to other articles published in this journal.

1) It is not clear when hair cell rearrangements occur relative to the formation and polarization of the stereociliary bundle. Are these concurrent or sequential events?

2) PCP signaling regulates a potentially analogous cellular rotation during ommatidia development in the *Drosophila* eye. PCP signaling is also essential in the neuromast to align the hair cell stereociliary bundles along a common axis. Thus it seems worthwhile to evaluate how (or if) the Rock and Roll dynamics are altered in PCP mutants.

3) The authors consistently use verbs such as “reverses” and “changes” to describe Emx2's function in establishing stereociliary bundle orientation in Emx2-positive hair cells. As written this suggests that the stereociliary bundles are actively rotating to assume an new orientation in this population of cells. However there is no evidence in the literature documenting rotation of the nascent bundle. Instead the bundle forms adjacent to the kinocilium and bundle orientation is established by translocation of the kinocilium to the opposite side of Emx2-positive cells that it does in Emx2-negative cells. This point may appear semantic but is actually quite important to keep clear in this context since the process described in the text is also a rotational event.

4) Sentence starting "Therefore we investigated our existing sc-RNAseq data…" contains a reference to Matern et al. As structured this sentence suggest that “our existing sc-RNAseq data” is that data previously published by Matern et al. yet there is no author overlap. I suspect that the reference is for the fish line and not the RNAseq data and this should be clarified.

Reviewer #3:

Ohta et al. describe roles for *emx2* in the development of hair cell polarity. They analyze the cell movements linked to the development of polar orientation in pairs of differentiating hair cells under both gain of function and loss of function conditions. They also describe an *emx2* GFP knock-in that they use to monitor onset of expression in the context of precursor division and cell rearrangement. In the absence of *emx2* function GFP+ cells still end up in the usual anterior position. The findings largely support analysis of Kozak et al., 2020, who observed that *emx2* transcripts are found in half of the hair cells in *emx2* mutants. The main difference in the models is that Kozak posits that *emx2* expression is initiated in the precursor and then downregulated in one daughter after division. The key finding in the current work is that *emx2*-driven GFP arises in only one daughter of the hair cell precursor: it is detectable in the posterior cell of the pair before the onset of repositioning (roll) movements or in the anterior cell when roll movements do not occur. It is therefore the initiation not the downregulation of *emx2* that results in asymmetry. To test whether the cell movements that promote asymmetry are altered, Ohta et al. examine changes in rearrangements in *emx2* GOF and LOF conditions. They find some differences in the frequency and duration of movements, but since cell positions are largely resolved conclude that *emx2* plays some role in regulating movements but is not necessary to do so. Rather it regulates bundle polarity in cells independently of cell movement.

1) The authors state that *emx2* does not determine the ultimate positions of hair cells. Indeed the loss of function data support these conclusions, that is GFP+ cells are all anterior in these mutants. However in GOF transgenics some GFP+ cells are now posterior, suggesting that *emx2* overexpression can indeed alter cell positioning. Is this correct and if so, it should be discussed.

2) The analysis of apical processes is unconvincing. The authors do not have the resolution to accurately measure individual apical processes. It therefore becomes difficult to know what the difference are with gain and loss of *emx2* expression. It is not clear that there are differences in morphology or in timing of protrusion. These conclusions would require additional measurements.

3) The conclusion based on scRNAseq data differs somewhat from that of Kozak et al. in terms of the relative timing of *atoh1a* and *emx2* expression. Pseudotime analysis is sensitive to a number of parameters that can give different results. A more nuanced analysis than a pseudotime heatmap is needed to resolve this issue, for example a differential expression test and associated statistics. Alternatively an analysis of expression in situ might address this question.

---

## [Author Response]

Reviewer #1:The manuscript by Ohta and colleagues deals with the role of the transcription factor Emx2 in organizing the bidirectional polarity of hair cells of the zebrafish neuromasts. Previous reports by the authors have identified Emx2 as responsible of the establishment of stereociliary bundle orientation in the vertebrate (mouse, chick, zebrafish) inner ear and zebrafish neuromasts. A recent report from Lopez-Schier lab suggests that the cell rearrangement process of nascent hair cell sibling cells to position them in the neuromast might be coupled with differential Emx2 expression to ensure their correct position within the neuromast. In this manuscript, the authors try to follow this question of whether Emx2 is required for sibling cell rearrangement and thus cell positioning or, alternatively, regulates bundle orientation as it does in the inner ear. For this, the authors generate a zebrafish reporter line of Emx2 to follow the expression of Emx2 during the rotation process and they also analyse cell rotation in Emx2 Lof and Gof embryos through life imaging. The authors conclude that while the rotation process is modified in Lof and Gof embryos (duration, phases), Emx2 does not impair the rotation process and thus not determine the position of sibling cells. Live imaging also allowed the authors to see differences in apical protrusions of Emx2-positive and Emx2-negative cells.The manuscript is well performed and digs into an interesting question but fails short in their analysis of Emx2 function and whether the rotation of siblings has any meaningful role.

Our additional results of nocodazole experiments and analyses of the vangl2 mutants showed that sibling rotation is important for putting the Emx2-positive HC to the anterior position. Although the experiments for identifying downstream targets of Emx2 in HCs are ongoing, we feel it is equally important to be able to exclude Emx2’s role in positional acquisition of HCs.

In general, the manuscript could improve the Introduction and Discussion text to discuss the different scenarios for emx2 role, role of the rotation, apical protrusion, downstream players of emx2 for bundle orientation.

Per reviewer’s suggestion and the fact that we have rearranged our figures, we have re-written the Introduction and part of the Discussion to tailor towards the additional results.

1) By live imaging, the authors describe the % of cells performing cell rearrangement ( 57%) and the percentage that do not (43%) and separate this process into a rock phase and a roll phase (inversion). The authors find that when Emx2 is not expressed (in -/-) or overexpressed, there are differences in the time spent in each phase but cell rearrangement still happens. One would assume that if emx2 differential expression makes cell´s to rearrange in order to segregate in space the ones positive Emx2 and the ones negative for Emx2 (as observed using the reporter line), no rearrangement would happen if the siblings are equal.The authors find that in Lof and Gof embryos, cell rearrangement takes place. As a control it is necessary to show that in these conditions all cells are either positive or negative for Emx2. If still minor differences of Emx2 expression between siblings exist, cell rearrangement might not be blocked for this reason. Is Emx2 overexpression clearly affecting all cells, and Lof? It would be good to have a picture showing Emx2 immunoreactivity or ISH in emx2 ^-/-^ neuromasts and in Gof embryos.In their previous published work (Jiang et al., 2017) they show Emx2 expression in Lof and Gof but at 4 dpf. In this case, I understand that the embryos are younger, at 2-3 dpf. Could the myo6 promoter driving Emx2 for Gof be milder at these stages?Most probably the conclusion that Emx2 does not directly regulate cell rearrangement is correct but this additional control would improve the manuscript.

We agree with the reviewer that asymmetry between the two sibling HCs, imparted by Notch signaling, is the underlying mechanism propelling the cell rearrangement. Even though Emx2 is one of the factors regulated by the Notch signaling pathway, our results showed that Emx2 itself, though facilitates the rearrangement process, is not the main determinant of this process.

Nevertheless, we agreed that including the immunostaining results for *Emx2* KO and GOF at 2 dpf is a good control (see Figure 4—figure supplement 1 and subsection “The HC rearrangement process is regulated by Emx2”). The hair bundle orientation phenotype in *Emx2* KO and F1 generation of the GOF neuromasts at 2 dpf are fully penetrant. The levels of Emx2 among GOF HCs are indeed different, as indicated by all or none endogenous Emx2 immunoreactivity between sibling HCs (anti-Emx2 staining) even though their ectopic Emx2 expression is ubiquitous (based on mCherry expression). Although we cannot preclude the possibility that the Rock and Roll in the *Emx2* GOF neuromast can be caused by the differential levels of Emx2 between two sibling HCs, it is very clear that Emx2 is not detected in the KO mutants at 2 dpf and HC rearrangement still occurs. Therefore, we think the differences in the levels of Emx2 between two sibling HCs is unlikely to drive the Rock and Roll process.

Another recent paper (Erzberger et al., 2020) has shown that cell rearrangement is impaired when siblings do not have differential activity of Notch pathway. Thus, being Emx2 downstream of Notch, the present results suggest that cell rearrangement mediated but Notch is driven by other downstream factors. The fact that probably Emx2 is not required for cell rearrangement but another factor is interesting.

We agree with the reviewer too. While Notch regulates Emx2 expression, which confers hair bundle orientation, other downstream factor(s) of Notch is mediating the HC rearrangement. Furthermore, our added *trilobite* mutant analyses indicate that the rearrangement is also dependent on intercellular signaling via the core PCP proteins (see results of Figure 9).

2) The analysis with the reporter indicates that GFP appears when cells perform cell rearrangement and is always correlated with the final position cells adopts, if emx2 positive are in posterior perform cell rearrangement, if are anterior located do not. In Gof, GFP cells can be found mislocated. This is a very nice result.

Thank you! The *Emx2* reporter fish that we generated allowed us to distinguish which HC within a pair is destined to be in the anterior position. Using this tool, we have further demonstrated that both disruption of microtubules and loss of cPCP signaling affect positional acquisition of nascent HCs (see Figures 7-9).

3) How the authors envisage that Emx2 regulates apical protrusions and cell shape changes? Can the authors affect apical protrusion formation to see if cell rotation is impaired?

The reviewer raised a good question. We addressed this question by blocking protrusion formation with transient nocodazole treatment, which blocks microtubule polymerization. Our results showed that introducing nocodazole immediately after the precursor cell divided caused the loss of the apical protrusion. Removal of nocodazole after 90 min of treatment, resulted in the recovery of protrusion after 12 min and this protrusion was followed by HC rearrangement. In contrast, HCs treated with prolong nocodazole for 150 min did not recover protrusion formation and consequently no protrusion was evident and some HCs were mispositioned. These results are consistent with the hypothesis that apical protrusion is important for the cell rearrangement process to occur (see Results section).

We don’t know how Emx2 regulates the apical protrusion. Based on existing evidence that Emx2 regulates symmetric and asymmetric cell divisions in the brain, basal body positioning in the HCs as well as delaying apical protrusion in nascent zebrafish HCs, we speculate that effectors of Emx2 may involve either centrioles/basal body or microtubules or both, since they seem to be the commonality among the effects of Emx2.

4) Previously, the authors indicated that in zebrafish utricule and cristae, emx2 is also involved in hair bundle polarity. I wonder if there a cell rearrangement between siblings also takes place as in the neuromasts. Live imaging in zebrafish inner ear could help to assess whether emx2 role in the rock and roll process and cell shape is specific of neuromasts organization or has a wider role.

The reviewer raised an interesting point. We now know that HC rearrangement observed in zebrafish neuromasts is important for HCs to acquire its proper location and to generate the bidirectional hair bundle pattern within the neuromast. Assuming the role of Emx2 in the utricle is conserved among fish, chicken and mice, our existing results in the mouse utricle indicates that HCs across the line of polarity reversal in the inner ear are born on different days during development (Jiang et al., 2017). Therefore, it is highly unlikely that HCs will rearrange across this border. Additionally, we also showed that *Emx2* is activated in the mouse utricle at least three days ahead of HC formation (Tona and Wu, *eLife* 2020). Therefore, if Emx2 has a similar role in establishing its domain via cell rearrangement in the utricle as in the neuromast, it is likely to occur earlier during patterning of the sensory epithelium rather than the HCs per se. To address this question, it will require a different set of tools and imaging techniques, which are beyond the scope of our current study.

Several manuscripts have been published in relation to the cell rearrangement process in the neuromasts and the mechanisms behind it, however, its biological implications if is very particular of this system, might be of low relevance. I think it is more relevant to explore deeply how emx2 changes bundle orientation.

Yes, HC rearrangement in neuromast has been described by two different labs 8 years ago, but the hypothesis that this process is directly involved in the positional acquisition of HCs has never been demonstrated. Here, using *emx2*-reporter line, we provided the proof that the HC rearrangement process is required for HCs to establish their positions. While our experiments in exploring the downstream effectors of Emx2 in mediating hair bundle orientation are ongoing, we felt it was equally important, if not more, to determine whether emx2 mediates HC positions, since positional acquisition precedes hair bundle establishment and specific HC positions alone could generate the bidirectional hair bundle pattern.

Furthermore, we respectfully disagree that if HC rearrangement is unique to the neuromast, it thus undermines the biological significance of the phenomenon. We feel any mechanism deployed in positional acquisition should be of importance in developmental biology since positional identity is fundamental to the organization and differentiation of most tissues. The zebrafish neuromast has a lot to offer as a model system in this regard since this organ is simple and easily accessible. It develops quickly and its functionally-relevant structure, the bidirectional HC orientation, is easily scorable.

5) The authors perform scRNA-seq to trace the emx2 expressing cells and establish the early expression of this transcription factor. Their analysis does not differ from the scRNAseq analysis and trajectory plot recently published in Kozak et al. and thus as it is presently, it does not add new information. I suggest eliminating this part or, if kept, use it to expand the analysis and perform functional experiments with putative downstream factors present in emx2 expressing cells that could regulate sterociliary bundle orientation. A deeper study of the implications of emx2 activity in bundle reorientation would make this manuscript more solid.

We apologize for not making the impetus and the implication of the seemingly subtle differences between our scRNAseq results and previously published results. We have now modified our Introduction and Discussion for clarification.

I think the reviewer would agree with us that pinpointing the onset of *Emx2* expression within the neuromast is an important piece of information for pursuing the role of Emx2 in establishing bidirectional HC pattern. Pinpointing the onset of Emx2 expression better than the immunostaining results was the reason for generating the *emx2:Gfp* reporter fish. Based on the *emx2-Gfp* reporter results, *emx2* is activated in one of the nascent HCs shortly after precursor cell division (Figure 3) and these results are consistent with the Rock and Roll phenotypes observed in the *emx2* mutants (Figure 4). Nevertheless, the recent paper by Kozak et al. (2020) earlier this year showed robust Emx2 immunostaining in the HC precursors. The onset of Emx2 expression in the precursor cells, if true, could totally change how Emx2 might be regulated in the HCs (see Discussion section) and also raise the question whether the emx2:Gfp reporter is sensitive enough.

Although neither we nor the Hudspeth’s lab (Jacobo et al., 2019) have observed Emx2 immunoreactivities in HC precursors using the same commercial source of antibody and the identity of the Emx2-positive precursor was not validated, Kozak et al. also showed the presence of *emx2* transcripts in the precursor cells using a sensitive in situ hybridization technique and scRNA seq approach. In the pursue of downstream effectors of Emx2, we had generated similar scRNA-seq results using 10x Genomics. While the hunt for downstream effectors of Emx2 is ongoing, results of Kozak et al. prompted us to analyze our scRNAseq results for onset of Emx2 expression by following closely the analysis conducted by Kozak et al. using SCANPY and the criteria they applied to their dataset. Our analyses showed that results of the two datasets are very similar except our results showed that the onset of *emx2* corresponds to the time when precursors transition to nascent HCs rather than being highly expressed in the precursor cells. This nascent HC population, however, is absent in the Kozak dataset. It is not clear whether the absence of nascent HC population in their dataset is due to age of the fish collected (7dpf in their study versus 4 dpf in ours) and/or the total numbers of HCs (1240 HCs in their study versus 4000 in ours), which will require further investigation. Nevertheless, our results of scRNA-seq, *emx2:Gfp* reporter activity and Emx2 mutant phenotypes are all consistent with each other supporting a role of Emx2 functioning at the nascent HC stage.

Reviewer #2:Here, the authors find that Emx2 regulates the development of aspects of hair cell morphology that influence rearrangements but does not appear to regulate the process directly. The work is largely descriptive and does not reveal a mechanistic or transcriptional link between Emx2 and hair cell morphology nor does it consider the contribution of other signaling pathways such as PCP signaling which also controls stereociliary bundle orientation in the neuromast. For these reasons it seems an minor advance and is currently not comparable to other articles published in this journal.

The research advance of this manuscript over the initial observation that emx2 is necessary and sufficient to establish the bidirectional pattern in zebrafish neuromast (Tao et al., *eLife* 2017) is to address how Emx2 generates the bidirectional orientation pattern in the neuromast. Since sibling HCs in the neuromast can undergo rearrangement before differentiating into HCs of opposite bundle orientation, it is highly possible that Emx2 may not be mediating hair bundle orientation in HCs but their positions within the neuromast. While searching for downstream effectors of Emx2 is ongoing, we felt this is an equally important question to tackle.

Furthermore, per reviewer’s request, we have now added additional results in this revision pertaining to the role of the core PCP pathway in HC rearrangement (see detail responses below).

1) It is not clear when hair cell rearrangements occur relative to the formation and polarization of the stereociliary bundle. Are these concurrent or sequential events?

Yes, these are sequential events. Nascent HCs undergo rearrangement shortly after the HC precursor divided but hair bundle formation is not apparent until HCs start to differentiate. We have revised our Introduction as well as Figure 1 to clarify this important point.

2) PCP signaling regulates a potentially analogous cellular rotation during ommatidia development in the *Drosophila* eye. PCP signaling is also essential in the neuromast to align the hair cell stereociliary bundles along a common axis. Thus it seems worthwhile to evaluate how (or if) the Rock and Roll dynamics are altered in PCP mutants.

The Rock and Roll dynamics in the *vangl* mutants, *trilobite*, has been described by the Hudspeth group (Mirkovic et al., 2012). Hair bundles are misoriented in *trilobite* neuromasts and the HC rearrangement process was described to stall or undergo multiple rolls, which extends the duration of the rearrangement. However, it was not clear whether the abnormal Rock and Roll described in these mutants directly caused HC mispositioning. Using *Emx2-Gfp* reporter fish, we showed that the positions of HCs after Rock and Roll were abnormal in *trilobite* (Figure 9), supporting the hypothesis that cell rearrangement is important for HCs to acquire their proper positions.

3) The authors consistently use verbs such as “reverses” and “changes” to describe Emx2's function in establishing stereociliary bundle orientation in Emx2-positive hair cells. As written this suggests that the stereociliary bundles are actively rotating to assume an new orientation in this population of cells. However there is no evidence in the literature documenting rotation of the nascent bundle. Instead the bundle forms adjacent to the kinocilium and bundle orientation is established by translocation of the kinocilium to the opposite side of Emx2-positive cells that it does in Emx2-negative cells. This point may appear semantic but is actually quite important to keep clear in this context since the process described in the text is also a rotational event.

We thank the reviewer for pointing out this oversight. The wording has been corrected in the revised manuscript.

4) Sentence starting "Therefore we investigated our existing sc-RNAseq data…" contains a reference to Matern et al. As structured this sentence suggest that “our existing sc-RNAseq data” is that data previously published by Matern et al. yet there is no author overlap. I suspect that the reference is for the fish line and not the RNAseq data and this should be clarified.

This sentence has been clarified.

Reviewer #3:Ohta et al. describe roles for emx2 in the development of hair cell polarity. They analyze the cell movements linked to the development of polar orientation in pairs of differentiating hair cells under both gain of function and loss of function conditions. They also describe an emx2 GFP knock-in that they use to monitor onset of expression in the context of precursor division and cell rearrangement. In the absence of emx2 function GFP+ cells still end up in the usual anterior position. The findings largely support analysis of Kozak et al., 2020, who observed that emx2 transcripts are found in half of the hair cells in emx2 mutants. The main difference in the models is that Kozak posits that emx2 expression is initiated in the precursor and then downregulated in one daughter after division. The key finding in the current work is that emx2-driven GFP arises in only one daughter of the hair cell precursor: it is detectable in the posterior cell of the pair before the onset of repositioning (roll) movements or in the anterior cell when roll movements do not occur. It is therefore the initiation not the downregulation of emx2 that results in asymmetry. To test whether the cell movements that promote asymmetry are altered, Ohta et al. examine changes in rearrangements in emx2 GOF and LOF conditions. They find some differences in the frequency and duration of movements, but since cell positions are largely resolved conclude that emx2 plays some role in regulating movements but is not necessary to do so. Rather it regulates bundle polarity in cells independently of cell movement.

We thank the reviewer for the concise summary of our results. Notably, the distinction between initiation versus downregulation of emx2, which we did not clarify well in the previous submission and is now better illustrated in the revised manuscript (see Figure 1C).

1) The authors state that emx2 does not determine the ultimate positions of hair cells. Indeed the loss of function data support these conclusions, that is GFP+ cells are all anterior in these mutants. However in GOF transgenics some GFP+ cells are now posterior, suggesting that emx2 overexpression can indeed alter cell positioning. Is this correct and if so, it should be discussed.

This point has been elaborated in the Discussion.

2) The analysis of apical processes is unconvincing. The authors do not have the resolution to accurately measure individual apical processes. It therefore becomes difficult to know what the difference are with gain and loss of emx2 expression. It is not clear that there are differences in morphology or in timing of protrusion. These conclusions would require additional measurements.

We have refined our 3-D rendering by increasing the number of the polygons generated for rendering. The results are improved (see Figure 6). Additionally, we have quantified the intensity of the GFP signals in individual apical protrusions using ImageJ Fiji plug in. This quantification and detailed description of the analysis are now included in Figure 6—figure supplement 1 and the Materials and method section.

3) The conclusion based on scRNAseq data differs somewhat from that of Kozak et al. in terms of the relative timing of atoh1a and emx2 expression. Pseudotime analysis is sensitive to a number of parameters that can give different results. A more nuanced analysis than a pseudotime heatmap is needed to resolve this issue, for example a differential expression test and associated statistics. Alternatively an analysis of expression in situ might address this question.

We agree with the reviewer that pseudotime analysis is sensitive to the parameters being applied, even though we have tried to adhere as closely to the clustering parameters described by Kozak et al. as possible. To strengthen our results, we compared the number of *emx2* read counts and percentages of Emx2-positive cells among the clusters. Our results showed that the nascent HC cluster showed significantly higher numbers for both parameters than other clusters, supporting our hypothesis that *emx2* transcripts are highest in the nascent HCs and not precursor cells.